# Stearic acid blunts growth-factor signaling via oleoylation of GNAI proteins

Hana Nůsková [1,2], Marina V. Serebryakova [3,5], Anna Ferrer-Caelles [1,2,5], Timo Sachsenheimer[4], Christian Lüchtenborg[4], Aubry K. Miller [1], Britta Brügger [4], Larisa V. Kordyukova [3] & Aurelio A. Teleman [1,2✉]

Covalent attachment of C16:0 to proteins (palmitoylation) regulates protein function. Proteins are also S-acylated by other fatty acids including C18:0. Whether protein acylation with different fatty acids has different functional outcomes is not well studied. We show here that C18:0 (stearate) and C18:1 (oleate) compete with C16:0 to S-acylate Cys3 of GNAI proteins. C18:0 becomes desaturated so that C18:0 and C18:1 both cause S-oleoylation of GNAI. Exposure of cells to C16:0 or C18:0 shifts GNAI acylation towards palmitoylation or oleoylation, respectively. Oleoylation causes GNAI proteins to shift out of cell membrane detergent-resistant fractions where they potentiate EGFR signaling. Consequently, exposure of cells to C18:0 reduces recruitment of Gab1 to EGFR and reduces AKT activation. This provides a molecular mechanism for the anti-tumor effects of C18:0, uncovers a mechanistic link how metabolites affect cell signaling, and provides evidence that the identity of the fatty acid acylating a protein can have functional consequences.

[1] German Cancer Research Center (DKFZ), Heidelberg, Germany. [2] Heidelberg University, Heidelberg, Germany. [3] A.N. Belozersky Institute of Physico-Chemical Biology, M.V. Lomonosov Moscow State University, Moscow, Russia. [4] Heidelberg University Biochemistry Center (BZH), Heidelberg, Germany. [5] These authors contributed equally: Marina V. Serebryakova, Anna Ferrer-Caelles. ✉email: a.teleman@dkfz.de

Cell signaling and cell metabolism are tightly interlinked. Signaling pathways can regulate cell metabolism (e.g., via post-translational modification of metabolic enzymes), and recent work has shown that the converse also occurs and cell metabolism affects cell signaling. This happens in different ways. For instance, metabolites can act as co-factors for signaling proteins (e.g., NAD$^+$ for sirtuins[1]) or as allosteric regulators[2,3]. Metabolites can also be enzymatically attached to proteins, thereby forming covalent post-translational modifications (PTMs) that directly regulate protein function. For instance, palmitoylation, acetylation, malonylation, and amino acylation are PTMs resulting from covalent attachment of cellular metabolites onto proteins. In some cases, metabolite levels affect the level of protein modification, as is the case for acetyl-CoA and protein acetylation. When acetyl-CoA levels are experimentally altered in cells in culture, this leads to correspondingly altered histone acetylation[3,4], thereby affecting cellular transcription. Hence the cellular levels of these metabolites can directly convert into the stoichiometry of protein post-translational modification, thereby linking cell metabolism to cell signaling.

One example of a PTM resulting from covalent attachment of a metabolite is S-acylation, which is the modification of a cysteine residue with a fatty acid via a thioester bond[5–7]. Since palmitoylation (C16:0) is the most common lipid modification, S-acylation was often referred to as palmitoylation in the past. It is now clear, however, that proteins can also be modified by other fatty acids such as stearate (C18:0) or oleate (C18:1)[8–11]. The presence or absence of S-acylation on proteins affects their localization, stability, and activity[5,7]. Indeed, the hydrophobicity of a protein changes dramatically when a fatty acid chain is either added or removed. Less clear, however, is whether the identity of the fatty acid being added to the protein is of functional relevance. A single given cysteine residue on a protein can be differentially acylated by distinct fatty acids[12]. Whether differential S-acylation of one residue leads to differential effects on protein function in animals is not well studied. Fatty acids such as C16:0 or C18:0 are similar enough biophysically that one could imagine the resulting S-acylations to be functionally equivalent. That said, heterogeneous S-acylation of Src family proteins regulates their raft localization[13], providing an example where differential S-acylation can affect protein function. This question is significant because it provides insight into whether the diversity of fatty acids that can be attached to proteins via S-acylation is of functional relevance, or whether different S-acylations are functionally redundant. Furthermore, this has implications for how metabolites can regulate cell signaling because different fatty acids are generated via distinct metabolic pathways.

We previously found that stearic acid, C18:0, can be covalently attached to proteins, and that the cellular and indeed organismal levels of C18:0 thereby affect cell signaling and physiology[14]. Exposure of cells to C18:0 in culture medium leads to covalent attachment of C18:0 to the transferrin receptor (TfR1), which activates a signaling pathway causing mitochondrial fusion[14]. This signaling pathway responds to organismal C18:0 levels. Flies lacking C18:0 have fragmented mitochondria and impaired respiration, whereas flies fed exogenous C18:0 have mitochondria that are more fused than usual[14]. We recently extended these findings to humans, where a double-blind cross-over clinical study showed that 3 h after we eat C18:0 but not C16:0 the mitochondria of neutrophils fuse and activate fatty acid beta-oxidation[15]. Altogether, these data indicate that cells in our bodies are poised to specifically sense dietary levels of C18:0, and to respond by altering cell signaling. This serves as a mechanism to couple levels of this dietary metabolite to cell signaling and organismal physiology.

Dietary fatty acids also modulate cancer growth. Although saturated fatty acids are generally harmful, the anti-cancer effects of a C18:0-rich diet have been well documented. High levels of dietary C18:0 cause decreased incidence and delayed development of spontaneous mammary tumors in mice[16,17] and inhibit chemically-induced tumorigenesis in rats[18,19]. Furthermore, a C18:0-rich diet reduces xenograft tumor size and tumor metastasis in mice[20,21]. In vitro, C18:0 inhibits cancer cell proliferation[22,23], colony formation[19], adhesion, and invasion[24,25]. The molecular mechanisms underlying these effects have not been fully elucidated. Part of these effects may be metabolic. Whether C18:0 affects oncogenic signaling via S-acylation is not known. Some data indicate that inhibition of the PI3K-AKT pathway is involved, since constitutively active AKT can overcome the suppressive effect of C18:0 on cell proliferation and tumor growth[22]. Furthermore, C18:0 was shown to inhibit EGF-stimulated proliferation of breast cancer cells[23].

A previous screen in our laboratory identified the inhibitory Gα-proteins GNAI1-3 as putatively S-acylated with C18:0[14]. GNAI proteins are membrane associated with two lipid modifications, N-myristoylation on Gly2 and S-acylation on Cys3. The fatty acid residue on Cys3 is thought to be mainly palmitate, C16:0[26,27]. GNAI1-3 are involved in two distinct sets of signaling pathways where they have opposite effects—in one case inhibiting signaling and in one case activating it. Downstream of G-protein coupled receptors (GPCRs), such as GABA$_B$ or the dopamine receptor, GNAI proteins inhibit adenylate cyclase and cAMP production[28–30]. Downstream of receptor tyrosine kinases (RTKs) such as EGFR, FGFR, VEGFR, and KGFR, however, GNAI proteins activate the PI3K-AKT-mTORC1 pathway[31–36]. The molecular details of how GNAI proteins enhance RTK signaling are not completely clear. According to one model, upon ligand binding, GNAI proteins bind to the activated RTK and to the GIV/Girdin guanine-nucleotide exchange factor (GEF). This activates the GNAI complex, causing the Gβγ subunits, as well as GIV/Girdin, to dissociate from the complex and to activate downstream PI3K[37–39]. According to another model, GNAI proteins help recruit the adaptor protein Gab1 (growth-factor receptor binding 2 (Grb2)-associated binding protein 1) to mediate subsequent activation of PI3K-AKT-mTORC1 signaling[31–33,40]. By modulating signaling through RTKs, GNAI proteins affect cellular growth, proliferation, and migration. For instance, KGF- and EGF-induced cell proliferation and migration are inhibited upon GNAI1/3 knockdown[31,33]. Consistent with this, GNAI1/3 expression is increased in both healthy and diseased cells that are proliferating, such as wounded human skin[33] or cancer cells[40]. In sum, the GNAI proteins play an important role in activating oncogenic signaling downstream of growth-factor receptors.

In this study, we uncover a molecular mechanism linking C18:0 and C18:1 to reduced EGFR signaling via GNAI proteins. We find that Cys3 of GNAI proteins can by modified by S-oleoylation, which is the covalent attachment of C18:1. We find that both C18:0 and C18:1 from cell culture medium are used by cells to generate S-oleoylation of GNAI proteins on Cys3. Since GNAI proteins can be modified with either C16:0 or C18:1 on this one residue, this reveals a competitive situation whereby GNAI proteins are either palmitoylated or oleoylated. We find that exposure of cells to C18:0 or C18:1 increases S-oleoylation of GNAI on Cys3, causing GNAI proteins to re-localize out of detergent-resistant membranes (DRMs) where EGFR is located. This blunts the ability of GNAI proteins to activate EGFR signaling via recruitment of the adapter Gab1. These findings indicate that C18:0 and C18:1 blunt oncogenic signaling downstream of EGFR via post-translational modification of GNAI

proteins. Furthermore, it reveals that S-acylation of a protein with different fatty acids has different functional consequences, identifying a mechanism how cell metabolism can regulate cell signaling.

## Results

**GNAI proteins are modified by both C15:0-azide and C17:0-azide in vivo.** To study protein S-acylation, we used an established metabolic labeling assay that employs fatty acid analogs containing an azide functional group at the methyl end (Fig. 1a)[41]. After these azido fatty acids are incubated with cells and incorporated into cellular material, the cells are lysed, and the azido fatty acid is conjugated to beads via a copper-catalyzed

azide-alkyne cycloaddition ("click chemistry") for subsequent isolation. Importantly, the resulting linkage is covalent, allowing stringent washing under denaturing conditions, to identify only proteins that are covalently attached to the lipids. To elute the proteins from the beads we use hydroxylamine, which hydrolyzes the thioester bonds diagnostic of S-acylation. The C16:0 analog C15:0-azide has been extensively used to identify palmitoylated proteins[42,43]. We previously showed that C17:0-azide is functionally equivalent to C18:0[14]. Using this assay followed by mass spectrometry, we previously screened for stearoylated proteins in HeLa cells[14]. Amongst the top hits were all three members of the GNAI family, GNAI1, GNAI2, and GNAI3. To validate these hits, and to check if GNAI proteins are also acylated in MCF7 cells, we

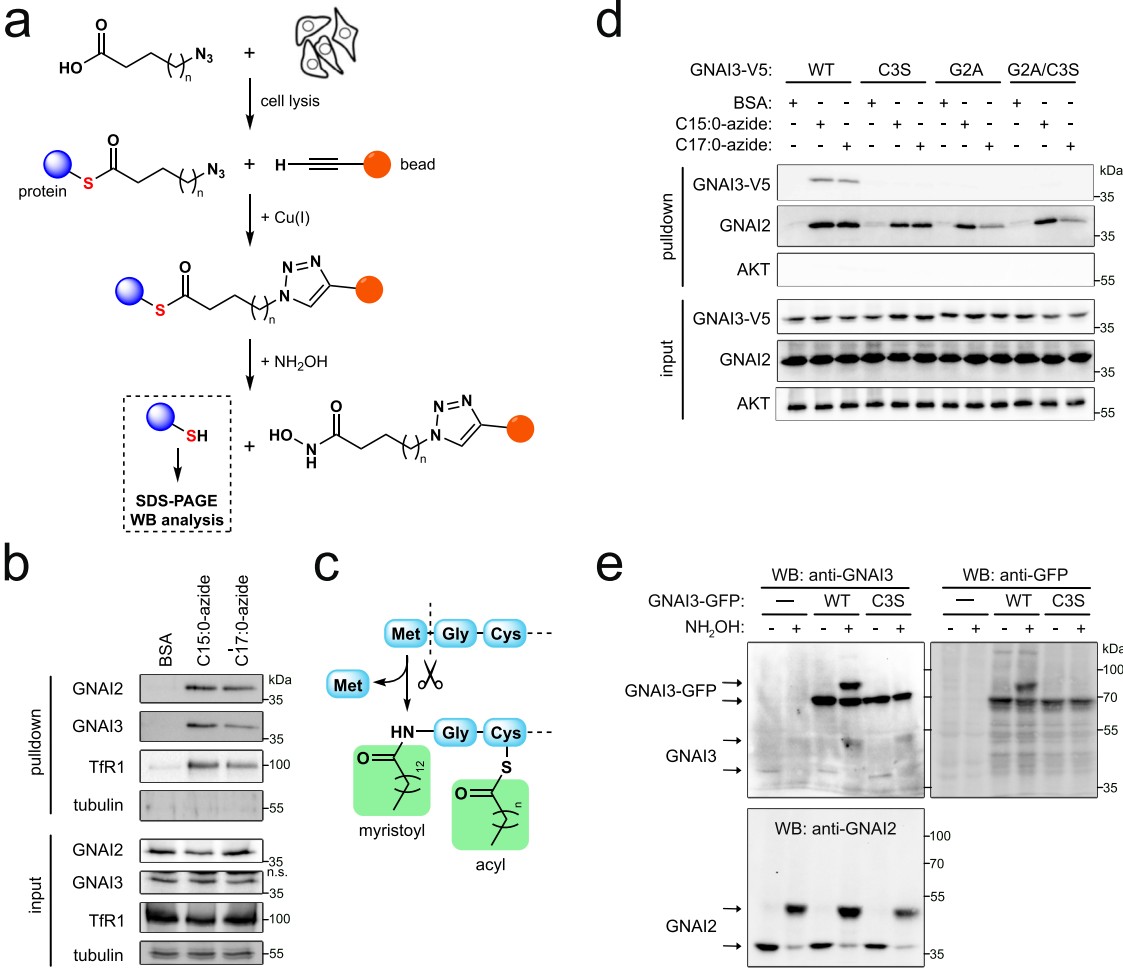

**Fig. 1 GNAI proteins are S-acylated by exogenous C15:0-azide and C17:0-azide. a** Schematic diagram of the detection of protein S-acylation using azido analogs of fatty acids, C15:0-azide ($n = 12$) and C17:0-azide ($n = 14$). Cells are metabolically labeled by incubating with BSA-conjugated azido fatty acids. After cell lysis, modified proteins are covalently coupled to alkyne beads and extensively washed in denaturing conditions. S-acylated proteins are eluted with hydroxylamine ($NH_2OH$), which specifically cleaves thioester bonds, and proteins in the eluate are detected by immunoblotting. **b** Endogenous GNAI2 and GNAI3 are covalently modified with azido analogs of palmitic acid (C15:0-azide) and stearic acid (C17:0-azide). MCF7 cells were incubated with 100 μM BSA-conjugated C15:0-azide or C17:0-azide in the culture medium for 3 h at 37 °C. Vehicle (BSA only) is used as a negative control. Covalently modified proteins were purified as shown in panel (**a**) and endogenous GNAI2 and GNAI3 proteins were detected by immunoblotting. Transferrin receptor (TfR1) serves as a positive control[14], and tubulin as a negative control. Representative of three biological replicates. **c** Diagram illustrating the N-terminus of GNAI proteins and the lipid modifications on Gly2 (N-myristoylation) and Cys3 (S-acylation). **d** GNAI3 is S-acylated on Cys3. The single mutant Cys3Ser (C3S) is neither acylated by C15:0-azide nor by C17:0-azide. As previously reported, co-translational myristoylation of Gly2 is required for subsequent S-acylation of Cys3, because both the single Gly2Ala (G2A) mutant and the double Gly2Ala/Cys3Ser (G2A/C3S) mutant are not acylated. Metabolic labeling with 100 μM azido fatty acids was performed as in panel (**a**). Endogenous GNAI2 is used as a positive control for the metabolic labeling. Representative of two biological replicates. **e** Endogenous GNAI2 and GNAI3 are S-acylated on only one site, assayed using the acyl-PEG exchange (APE) assay. After blocking all free cysteine residues with N-ethylmaleimide, the thioester bonds of S-acyl-cysteines are cleaved by hydroxylamine ($NH_2OH$), releasing free cysteines which are then labeled with mPEG, yielding a mass shift corresponding to one modification. Mutation of Cys3 to Ser on GNAI3-GFP confirms that Cys3 is the only residue modified by S-acylation. Representative of three biological replicates.

metabolically labeled MCF7 cells with the C16:0 and C18:0 analogs and then detected the purified proteins by immuno-blotting. For this purpose, we first validated the GNAI antibodies. Overexpression studies revealed that anti-GNAI2 antibody can detect GNAI2 whereas anti-GNAI1 and anti-GNAI3 antibodies cross-react to detect both GNAI1 and GNAI3 due to their high homology (Supplementary Fig. 1a). Loss-of-function assays showed that the GNAI2 antibody detects GNAI2 whereas both GNAI1 and GNAI3 antibodies detect endogenous GNAI3 (Supplementary Fig. 1b, c) consistent with MCF7 cells expressing low levels of GNAI1 (Supplementary Fig. 1d). Therefore, throughout this paper, we refer to the band detected by GNAI1 and GNAI3 antibodies as GNAI3. Using these antibodies, we found that endogenous GNAI2 and GNAI3 can be S-acylated by both the palmitic acid analog C15:0-azide and the stearic acid analog C17:0-azide (Fig. 1b). This is consistent with previous studies showing that the GNAI proteins are palmitoylated[26,27]. To our knowledge, however, the modification of GNAI proteins by a C18:0-derived lipid has not yet been reported.

**GNAI proteins are S-acylated on Cys3**. We next asked which amino acid is modified by C17:0-azide. The N-terminus is conserved across the three GNAI proteins, and its lipid modifications have been studied in detail[26,27,44]. The N-terminal glycine (Gly2) is myristoylated co-translationally through an amide linkage (Fig. 1c). This modification is an essential prerequisite for post-translational palmitoylation on the neighboring cysteine (Cys3). Both lipid modifications are required for plasma membrane localization of GNAI1[26]. We therefore hypothesized that GNAI proteins can be modified by either palmitate or stearate on Cys3. Indeed, mutating Cys3 to serine abolished the modification of GNAI3 by either C15:0-azide or C17:0-azide (Fig. 1d). Modification of endogenous GNAI2 was used as a positive control that the metabolic labeling worked properly (Fig. 1d). Consistent with myristoylation of Gly2 being required for acylation of the neighboring cysteine[45], the Gly2Ala mutant could not be acylated by either C15:0-azide or C17:0-azide. We confirmed that Cys3 is the only S-acylated residue in GNAI3 using the acyl-PEG exchange (APE) assay, whereby S-acyl groups are cleaved from cysteines with hydroxylamine ($NH_2OH$), and the resulting free cysteines are labeled with a PEG mass-tag to induce a shift on an SDS-PAGE gel[46]. Although this method does not resolve the identity of the fatty acids, it showed that GFP-tagged GNAI3 (WT) is S-acylated, whereas GNAI3(Cys3Ser)-GFP is not (Fig. 1e), indicating that Cys3 is the only S-acylated residue. In sum, Cys3 of GNAI3 is modified either with C15:0-azide or with C17:0-azide, indicating that in cells palmitate and stearate compete for acylation of GNAI3.

**Mass spectrometry reveals Cys3 is either palmitoylated or oleoylated**. Azido fatty acids can be partially metabolized (e.g., desaturated) after they are taken up by cells[11]. Hence to confirm the identity of the endogenous lipid modifications on GNAI proteins, we expressed and purified GNAI3 with a C-terminal GFP-tag and directly analyzed the protein with its attached fatty acids by Matrix-Assisted Laser Desorption/Ionization Time-Of-Flight (MALDI-TOF) mass spectrometry (MS). Worth mentioning is that we noticed that S-acylation of GNAI proteins is very unstable in cell lysates, and it can be preserved by including palmostatin B, a specific inhibitor of acyl-protein thioesterases (APTs), in the lysis buffer at the time of cell lysis (Fig. 2a, Supplementary Fig. 2a). This enabled us to detect the endogenous acylation of GNAI3-GFP by MS, where we identified 3 major types of lipid modifications: N-myristoylation (C14:0) on Gly2, S-palmitoylation (C16:0) on Cys3, and S-oleoylation (C18:1) on

Cys3 (Fig. 2b, c, Supplementary Fig. 2b). A small amount of S-palmitoleate (C16:1) attached to Cys3 is also visible. Since we see that the azido analog of C18:0 is covalently attached to GNAI proteins on Cys3 (Fig. 1b–d) but we detect by mass spectrometry C18:1 as the dominant C18 modification on Cys3, this finding suggests that C18:0 becomes desaturated in the cell, either before or after it is covalently attached to the GNAI proteins. To test this hypothesis, we performed a tracer experiment whereby we fed cells heavy-carbon labeled stearate ($^{13}$C18:0) for 3 h and then analyzed the acylation of GNAI3-GFP by MS. This analysis yielded two results. First, we detected heavy $^{13}$C18:1 attached to Cys3 (Fig. 2d), confirming that C18:0 from the culture medium becomes C18:1 on GNAI3. Second, exposure of cells to heavy C18:0 increased the proportion of Cys3 modified with C18:1 versus C16:0, indicating that the balance of GNAI palmitoylation versus oleoylation is influenced by the lipid environment of the cell. To analyze whether C18:0 is desaturated to C18:1 before or after it is attached to GNAI proteins, we tested the effect of the desaturase inhibitor MF-438[47]. This revealed that in the presence of MF-438, heavy C18:0 becomes the predominant C18 species on GNAI3, rather than C18:1 (Fig. 2e), indicating that C18:0 can be used by cells to acylate GNAI3. To test the converse, we added heavy-carbon labeled C18:1 to cells, and found that it can also be used to acylate GNAI3 (Fig. 2f). In sum, both C18:0 and C18:1 can be covalently attached to GNAI3 on Cys3. Since MF-438, an acyl-CoA desaturase inhibitor, strongly reduced the amount of C18:1 acylation, this indicates that in vivo C18:0 is mostly first desaturated and then attached to GNAI. Altogether, these MS data show that C18-lipids directly modify GNAI3 resulting in oleoylation, and confirms that palmitoylation and oleoylation of GNAI3 are mutually exclusive and competitive, since they occur on the same amino acid residue.

We noticed that acylation of Cys3 happens quite rapidly. Only 3 h after adding $^{13}$C18:0 to cells, the MS peak corresponding to the peptide with $^{13}$C18:1 ($m/z = 1942.2$) is as large as the peaks for the palmitoylated peptide ($m/z = 1898.1$) or the non-acylated peptide ($m/z = 1716.9$) (Fig. 2d). This indicates that roughly half of the protein becomes newly acylated within 3 h (given that new acylation can also be with C16:0). This could either be due to S-acylation of newly synthesized protein, or to S-acylation of the existing pool of GNAI protein. To distinguish these two options, we analyzed GNAI3 half-life using cycloheximide (CHX) to block protein synthesis. c-Myc, which has a half-life of circa 30 min, served as a positive control, and was strongly diminished within 1 h of CHX treatment (Fig. 2g). In contrast, GNAI proteins are stable, with half-lives of >12 h (Fig. 2g–g′). This strongly suggests that GNAI proteins can be de-acylated and re-acylated in cells.

**The proportion of GNAI Cys3 that is oleoylated versus palmitoylated shifts upon exposure of cells to different lipids**. The acylation of GNAI3 Cys3 described above is an example of a post-translational modification (PTM) that results from the covalent attachment of a metabolite, such as palmitate or oleate, to a protein. In some cases, the stoichiometry of such PTMs depends on the activity of the enzymes that catalyze the modifications, whereas in other cases it depends on the abundance of the corresponding metabolites[48]. For instance, the amount of protein phosphorylation depends on the activity of kinases and not cellular ATP concentration, whereas protein acetylation is strongly influenced by levels of acetyl-CoA in the cell[48]. We therefore asked whether the differential acylation of GNAI Cys3 with either palmitate or oleate is affected by activity of the acyltransferases or by exposure of cells to different lipids. S-acylation is mediated by the zinc-finger DHHC-domain containing (ZDHHC) family of protein acyltransferases[5,6,49]. Recent studies have shown that

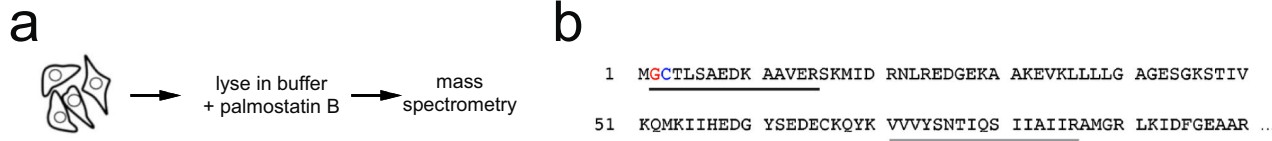

many members of the ZDHHC family have the potential to use both C16:0 and C18:0 as substrates[11,50]. This raises the possibility that the selectivity of S-acylation does not depend on the enzymes but rather on the relative abundance of the acyl-CoA substrates. A previous study screened all ZDHHC enzymes for their ability to acylate GNAI proteins, and found that GNAI proteins are mainly acylated by ZDHHC3 and ZDHHC7[51]. We therefore tested the contributions of ZDHHC3 and ZDHHC7 toward GNAI2 and GNAI3 palmitoylation and oleoylation in vivo using the metabolic labeling assay with azido fatty acids. Knockdown of ZDHHC3 had little effect on the labeling of GNAI2 or GNAI3

with either C15:0-azide or C17:0-azide, whereas knockdown of ZDHHC7 mildly reduced both (Fig. 3a and Supplementary Fig. 2c). This is consistent with a previous report that ZDHHC7 can use C16:0 and C18:0 almost equally well as substrates[11]. Combined knockdown of ZDHHC3 and ZDHHC7 did not further reduce acylation of GNAI2 or GNAI3 compared to knockdown of ZDHHC7 alone, suggesting that multiple enzymes act redundantly to acylate GNAI proteins in vivo (Fig. 3b and Supplementary Fig. 2d). Consistent with the fact that ZDHHC7 can use both palmitate and stearate as substrates, overexpression of ZDHHC7 did not shift the balance of S-acylation of GNAI

**Fig. 2 GNAI proteins can be either palmitoylated or oleoylated on Cys3. a–f** Detection of endogenous acylation of GNAI3-GFP by mass spectrometry. **a** Cells were lysed in buffer containing palmostatin B (250 μM), then GNAI3-GFP was purified and analyzed by MALDI-TOF mass spectrometry. **b** The N-terminal sequence of GNAI3 is shown, indicating the differentially acylated peptide (amino acids [2–15], underlined in black) and an inner hydrophobic peptide (amino acids [71–86], underlined in gray) detected in the mass spectrum region shown in panel (**c**). **c–f** Mass spectra of GNAI3-GFP peptides from GNAI3-GFP expressing cells that were either left untreated (**c**, "control"), or fed $^{13}$C-labeled C18:0 (**d**, "+$^{13}$C18:0"), $^{13}$C-labeled C18:0 in the presence of MF-438 (5 μM) (**e**, "+$^{13}$C18:0 +MF-438"), or $^{13}$C-labeled C18:1 (**f**, "+$^{13}$C18:1") for 3 h (100 μM BSA-conjugated fatty acids). For each condition, the whole MS spectrum is shown on the left and the region containing the N-terminal peptide in detail on the right. The $m/z$ values, the first and the last amino acids of the identified peptides, and their fatty acid modifications are indicated above the corresponding peaks. GNAI3-GFP was alkylated with IAA during sample preparation to block free cysteines yielding carbamidomethyl-Cys (cbm). A new wide peak was detected in the spectrum obtained with the $^{13}$C18:0-labeled sample, corresponding to addition of $^{13}$C-oleate (C18:1) (predominant peak at $m/z$ 1942) and $^{13}$C-stearate (C18:0) (shoulder at $m/z$ 1944). **g–g'** GNAI proteins are very stable, with half-lives >12 h, determined with a cycloheximide time course. **g** Cells were treated with 100 μg/mL cycloheximide (CHX) for indicated times. Unlike c-Myc, which degrades almost entirely within 1 h, GNAI proteins are stable for many hours. Quantified in (**g'**): For each timepoint, protein levels in the CHX-treated sample are shown relative to their respective DMSO-treated controls, following normalization of all data to tubulin whose levels do not change within 24 h. Data represent mean ± SEM of three biological replicates.

proteins toward either palmitoylation or oleoylation (Fig. 3c). Interestingly, ZDHHC7 itself was bound to both C15:0-azide and C17:0-azide (Fig. 3c), as expected from the fact that ZDHHC acyltransferases acylate target proteins via a 2-step relay mechanism involving first a transient auto-acylation on an active-site cysteine[52], and confirming that ZDHHC7 can use both C16:0 and C18:0 as substrates. Indeed, most of this signal was gone when the active-site Cys160 on ZDHHC7 was mutated to alanine (Fig. 3d). Furthermore, overexpression of *ZDHHC7* did not increase the total level of GNAI acylation (Fig. 3c), indicating that ZDHHC activity is not limiting for GNAI acylation in vivo. In sum, neither overexpression nor knockdown of individual acyltransferases had much of an effect on GNAI acylation, suggesting that the acyltransferases are acting in a non-limiting and largely redundant fashion.

In contrast, exposure of cells to different lipids in the medium strongly influenced GNAI acylation. This can be seen in Fig. 2c–f where we exposed cells to heavy-labeled C18:0 or C18:1 and found large increases in the corresponding peaks on Cys3 relative to the C16:0 peak by mass spectrometry. Likewise, exposure of cells to unlabeled C16:0 or unlabeled C18:0 for 3 h shifted the ratio of Cys3 acylation toward palmitoylation or stearoylation/oleoylation, respectively (Fig. 3e). This would suggest that differential acylation of GNAI proteins is driven by changes in the levels of the acyl-CoA substrates used by acyltransferases. Indeed, exposure of cells to C16:0, C18:0, or C18:1 caused an increase in the corresponding intracellular acyl-CoA species (Fig. 3f). Interestingly, C18:0 in the medium increased both intracellular C18:0-CoA and C18:1-CoA levels. In fact, the absolute increase in C18:1-CoA was larger than the absolute increase in C18:0-CoA (0.73 and 0.19 fmol per pmol of total phosphatidyl cholines, respectively). This agrees with the MF-438 result above indicating that C18:0 is mainly first desaturated to C18:1 and then attached to GNAI proteins. In sum, the relative stoichiometry of GNAI palmitoylation versus oleoylation reflects the relative abundance of the lipids to which cells are exposed. This agrees with findings from our previous clinical study whereby ingestion of C18:0 caused activation of a signaling pathway involving TfR1 stearoylation within 3 h in vivo in humans[15].

**Exposure of cells to stearic acid shifts GNAI proteins out of detergent-resistant membranes**. Acylation of G-proteins can affect their subcellular localization[13,53,54]. Hence to test the functional consequences of GNAI oleoylation we added C16:0 or C18:0 to the cell culture medium to skew GNAI acylation and asked whether this causes the localization of GNAI proteins to change. In agreement with previous work[55,56], acylation of the GNAI proteins is required for membrane localization because

mutants of GNAI1/2/3 that lose either one (Cys3Ser mutant) or both lipid modifications (Gly2Ala mutant and Gly2Ala/Cys3Ser double mutant) become cytosolic (Fig. 4a and Supplementary Fig. 3a, b). (The fact that overexpressed WT GNAI protein localizes correctly to the plasma membrane indicates that Gβγ subunits are not limiting in these cells, since association with prenylated Gβγ subunits is required for membrane targeting Fig. 4a[57].) Treatment of MCF7 cells with either C16:0 or C18:0, however, did not detectably alter the localization of endogenous GNAI2/3 to the cell membrane, detected either by immuno-fluorescence (Fig. 4b, Supplementary Fig. 4a), or by sucrose fractionation (Supplementary Fig. 4b). The plasma membrane is not homogenous—it has cholesterol and sphingolipid-rich domains where many signaling interactions occur[58], which can be detected as Detergent-Resistant Membranes (DRMs). Indeed, both GNAI proteins as well as the EGF receptor localize to DRMs[55,59–63], suggesting that DRMs are the site where GNAI proteins physically associate with EGFR to activate downstream signaling. We therefore tested whether treatment of cells with C18:0 alters localization of GNAI proteins to DRMs. DRMs can be separated from other cellular membranes by solubilizing cells with non-ionic detergent at 4 °C and separating fractions by density centrifugation with iodoxanol (OptiPrep)[58]. Most membrane proteins are found at the bottom of the gradient, whereas DRMs float toward the top and are positive for markers such as flotillin or caveolin. In agreement with published reports, we observed a portion of GNAI2 and GNAI3 localized to DRMs (see control sample treated with BSA as vehicle, Fig. 4c). Treatment of cells with C18:0, but not C16:0, reduced the amount of GNAI2 and GNAI3 in DRMs ("C18:0", Fig. 4c, quantified in Fig. 4d). This is in line with the fact that free C16:0 has the biochemical property of partitioning to DRMs, whereas C18:1 segregates out of DRMs[64]. In sum, these data indicate that GNAI proteins are modified on Cys3 with either C16:0 or C18:1, and this regulates partitioning of GNAI proteins in or out of DRMs (Fig. 4e).

**C18:0 reduces recruitment of Gab1 to EGFR via GNAI proteins**. EGFR activates AKT signaling mainly via recruitment of the docking protein Gab1, which in turn recruits PI3K (Fig. 5a)[65]. According to one mechanistic model, GNAI proteins help recruit Gab1 to EGFR thereby potentiating EGFR signaling[31–33,40]. In agreement with this, pulldown of EGFR-GFP revealed increased Gab1 binding upon EGF stimulation (lanes 3 and 4, Fig. 5b) and this was reduced in *GNAI3* knockout cells (lane 8 vs 4, Fig. 5b, and Supplementary Fig. 5a). Note that *GNAI3* knockout cells transcriptionally elevate *GNAI1* expression (Supplementary Fig. 5b), such that GNAI1 protein is now detectable in GNAI3 KO cells (the remaining band in Fig. 5b). Since exposure of cells to C18:0 shifts GNAI proteins out of DRMs and hence away from

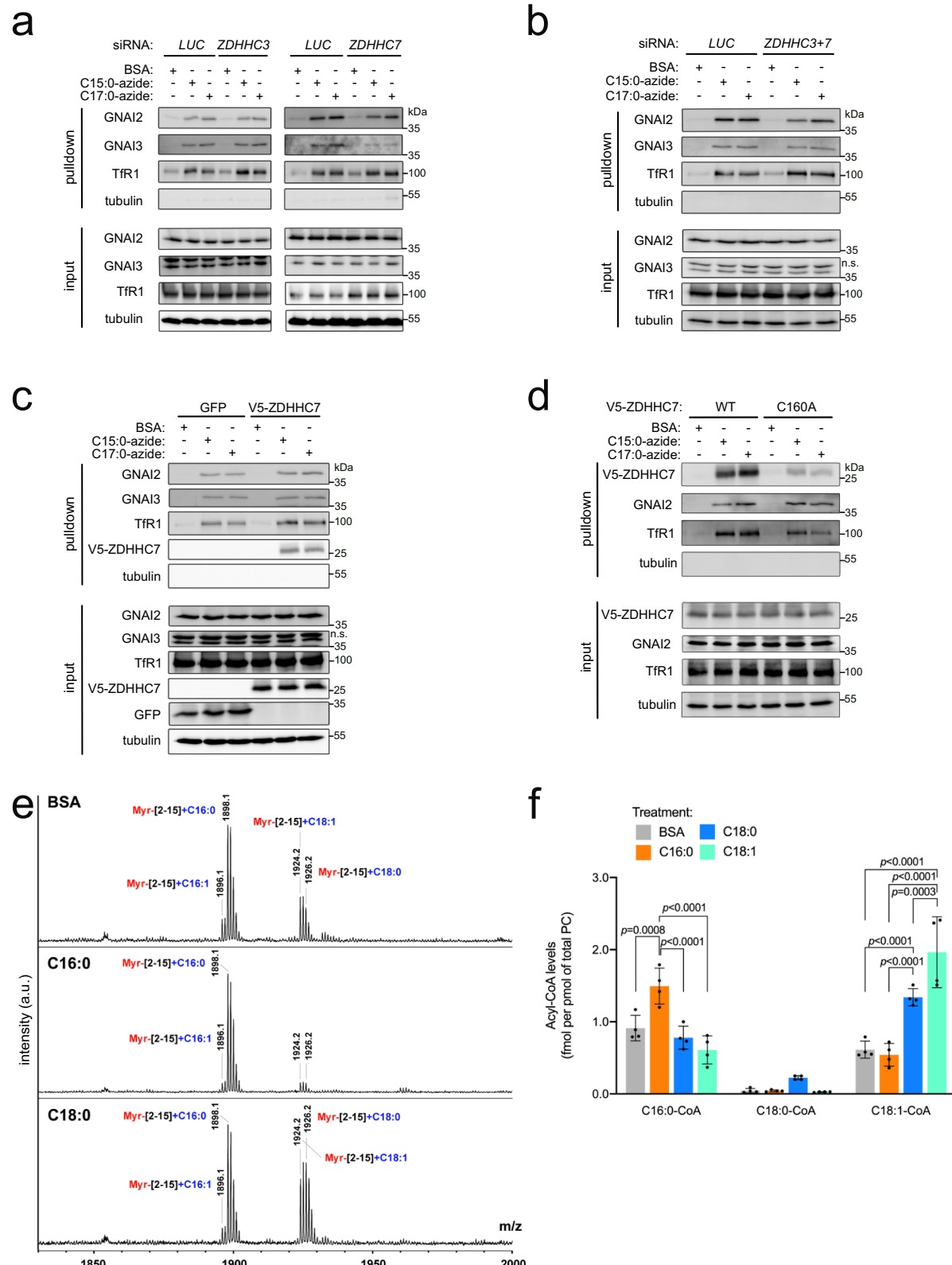

EGFR, this raises the possibility that C18:0 inhibits the ability of GNAIs to potentiate EGFR signaling. Consistent with this, treatment of cells with either C18:0 or C18:1 for 3 h reduced Gab1 recruitment to EGFR (Fig. 5c). After 24 h of treatment with lipids, C18:0 still reduced Gab1 recruitment (Supplementary Fig. 5d) whereas C18:1 did not (Supplementary Fig. 5e), perhaps because C18:1 is metabolized more rapidly[66]. To test whether C18:0

affects EGFR signaling via GNAI proteins, we performed two epistasis experiments. First, we combined a *GNAI3* knockout with C18:0 treatment. This revealed that C18:0 treatment reduces Gab1 recruitment in wild-type cells but not in *GNAI3* knockout cells where Gab1 recruitment is constitutively low (Supplementary Fig. 5c). This indicates that C18:0 requires the GNAI proteins to act on EGFR. Second, we overexpressed GNAI3 to see if this

**Fig. 3 The stoichiometry of GNAI3 palmitoylation versus oleoylation depends on the levels of the lipids the cells are exposed to. a, b** GNAI2 and GNAI3 are S-acylated with C15:0- and C17:0-azide by multiple acyltransferase enzymes in a redundant fashion. GNAI2 and GNAI3 acylation are only mildly reduced upon siRNA-mediated knockdown of *ZDHHC7* (**a**), and still acylated upon combined knockdown of *ZDHHC3* and *ZDHHC7* (**b**). Metabolic labeling with 100 µM azido fatty acids was performed as in Fig. 1. Representative of three biological replicates. **c** Overexpression of *V5-ZDHHC7* does not shift the balance in S-acylation of GNAI2 and GNAI3 proteins with C15:0- versus C17:0-azide. Metabolic labeling with 100 µM azido fatty acids was performed as in Fig. 1. Representative of four biological replicates. **d** V5-ZDHHC7 is modified by both C15:0- and C17:0-azide. Mutation of the active-site cysteine (C160A) blunts this acylation. Metabolic labeling with 100 µM azido fatty acids was performed as in Fig. 1. Representative of two biological replicates. **e** Exposure of cells to either C16:0 or C18:0 in the medium increases the fraction of GNAI protein that is either palmitoylated or oleoylated, respectively. Endogenous S-acylation of GNAI3-GFP was analyzed as in Fig. 2c in cells treated with different unlabeled fatty acids (100 µM for 3 h). **f** Exposure of cells to different fatty acids in the medium (100 µM for 3 h) shifts the intracellular pool of acyl-CoAs toward the corresponding fatty acid. Acyl-CoA levels were normalized to the total phosphatidyl choline (PC) content. Data are presented as mean ± SD of four biological replicates and the *p*-values were calculated by two-way ANOVA followed by Tukey's post hoc test.

rescues the effect of C18:0. If C18:0 treatment shifts part of the GNAI protein pool out of DRMs, overexpression of GNAI3 should compensate for this, allowing enough GNAI3 to reside in DRMs and to activate EGFR signaling. Indeed, this was the case (Fig. 5d). Together, these epistasis experiments indicate that C18:0 reduces EGFR signaling via the GNAI proteins.

**Oleoylation of GNAI proteins upon cell exposure to stearate suppresses AKT activation in response to EGF.** Recruitment of Gab1 to EGFR leads to activation of PI3K and phosphorylation of AKT[31–36]. Therefore, we asked whether AKT phosphorylation is affected by C18:0. We first confirmed that knockdown of *GNAI1* or *GNAI3* impairs the ability of EGF to activate AKT, assayed via phosphorylation of its two main regulatory sites Thr308 and Ser473 (compare lanes 5 and 11 to lane 2, Fig. 6a, quantified in Fig. 6a′). In contrast, and in agreement with previous reports[31,33,34], knockdown of *GNAI2* did not strongly blunt AKT phosphorylation in response to EGF in MCF7 cells (lane 8, Fig. 6a–a′), and knockdown of *GNAI1, 2,* or *3* had little effect on the stimulation of AKT phosphorylation in response to insulin (Fig. 6a). Conversely, overexpression of wild-type *GNAI3* enhanced AKT phosphorylation in response to EGF (lane 5 vs 2, Fig. 6b–b′). This potentiation of AKT phosphorylation was dependent on GNAI3 acylation, because it did not occur upon overexpression of mutant GNAI3 lacking the acylation sites (Fig. 6b–b′).

We next tested the functional consequences of exposing cells to increased levels of C16:0 or C18:0. In agreement with C18:0 reducing the amount of GNAI proteins in DRMs (Fig. 4c, d), treatment of cells with C18:0 blunted the phosphorylation of AKT in response to EGF stimulation (Fig. 6c–c′). Likewise, treatment of cells with C17:0-azide also blunted AKT phosphorylation in response to EGF (Supplementary Fig. 6a–a′), confirming that C17:0-azide is functionally equivalent to C18:0[14]. We reasoned that if C18:0 is causing part of the GNAI pool to shift out of DRMs, thereby reducing the amount of GNAI protein available to potentiate EGFR signaling, this should be rescued by over-expressing GNAI protein. Indeed, overexpression of *GNAI3-GFP* rescued the effect of C18:0 and caused constitutively high AKT phosphorylation in response to EGF (Fig. 6d–d′), showing that the effect of C18:0 on AKT is via GNAI proteins.

**Cell exposure to C18:0 inhibits cell proliferation.** AKT activation downstream of EGFR promotes cell proliferation[67,68]. Hence, from the data presented above, we would expect that loss of GNAI proteins or exposure of cells to C18:0 should reduce cell proliferation. Indeed, this is the case. Knockdown or knockout of *GNAI1* or *GNAI3* reduces MCF7 proliferation (Supplementary Fig. 7a, b). Likewise, treatment of MCF7 cells with C18:0 also causes reduced proliferation (Supplementary Fig. 7c). The

presence of 100 µM C18:0 in the medium increases the doubling time of MCF7 cells by 2.5 h (from 24.7 h for control BSA-treated cells or 24.6 h for C16:0 treated cells to 27.1 h for C18:0 treated cells). Since exposure of cells to C18:0 leads to acylation of several different proteins[14], the effect of C18:0 on cell proliferation is likely in part via GNAI proteins and in part via other mechanisms.

**Discussion**

Proteins can be acylated by different fatty acids, including C14:0, C16:0, C18:0, C18:1, C18:2, and C20:4[9,11]. It is unclear whether acylation by each of these fatty acids is functionally redundant in animals or not. We show here that acylation of Cys3 of GNAI proteins with either C16:0 or C18:1 leads to functional differences. Acylation with C16:0 leads to localization of GNAI proteins into detergent-resistant fractions of the cell membrane, whereas acylation with C18:1 does not. This has functional consequences for the ability of GNAI proteins to potentiate EGFR signaling and hence oncogenic signaling in cells. These results are consistent with a proof-of-concept study in which recombinant GNAI proteins were purified from *E. coli*, acylated chemically in vitro, and loaded onto sphingolipid- and cholesterol-rich liposomes to find that acylation with saturated fatty acids leads to partitioning into detergent-resistant fractions whereas acylation with unsaturated fatty acids leads to exclusion from DRMs[56]. We also observed a small amount of GNAI acylation with C16:1 (Fig. 2c). Although this may also shift GNAI proteins out of DRMs, the functional significance of this requires further study. A previous study showed that differential acylation of Src family proteins with saturated or unsaturated fatty acids also causes them to shift in or out of DRMs, respectively[13], indicating this may be a general mechanism regulating the function of DRM-localized S-acylated proteins. A previous study found that GNAI proteins can be acylated by ZDHHC3 and ZDHHC7 when co-transfected in HEK293T cells, with a smaller contribution from ZDHHC2 and ZDHHC21[51]. The fact that knockdown of *ZDHHC3* and *7* only had a minor impact on GNAI acylation (Fig. 3) suggests that endogenous ZDHHC2 and ZDHHC21 may also contribute to GNAI acylation in vivo, and that multiple enzymes are acting redundantly in this regard. Altogether, our data suggest that the relative acylation of GNAI3 Cys3 with C16:0 or C18:1 does not depend on changes in enzyme activity, but reflects instead the balance in fatty acids to which a cell is exposed.

The data we present here are consistent with a mechanistic model whereby exposure of cells to C18:0 or C18:1 leads to GNAI S-oleoylation on Cys3, in a manner that is competitive with palmitoylation with C16:0. This leads to exclusion of GNAI proteins from DRMs, where EGFR is present, thereby reducing the ability of the GNAI proteins to transduce the EGFR signal to AKT (Supplementary Fig. 8). This can potentially explain some

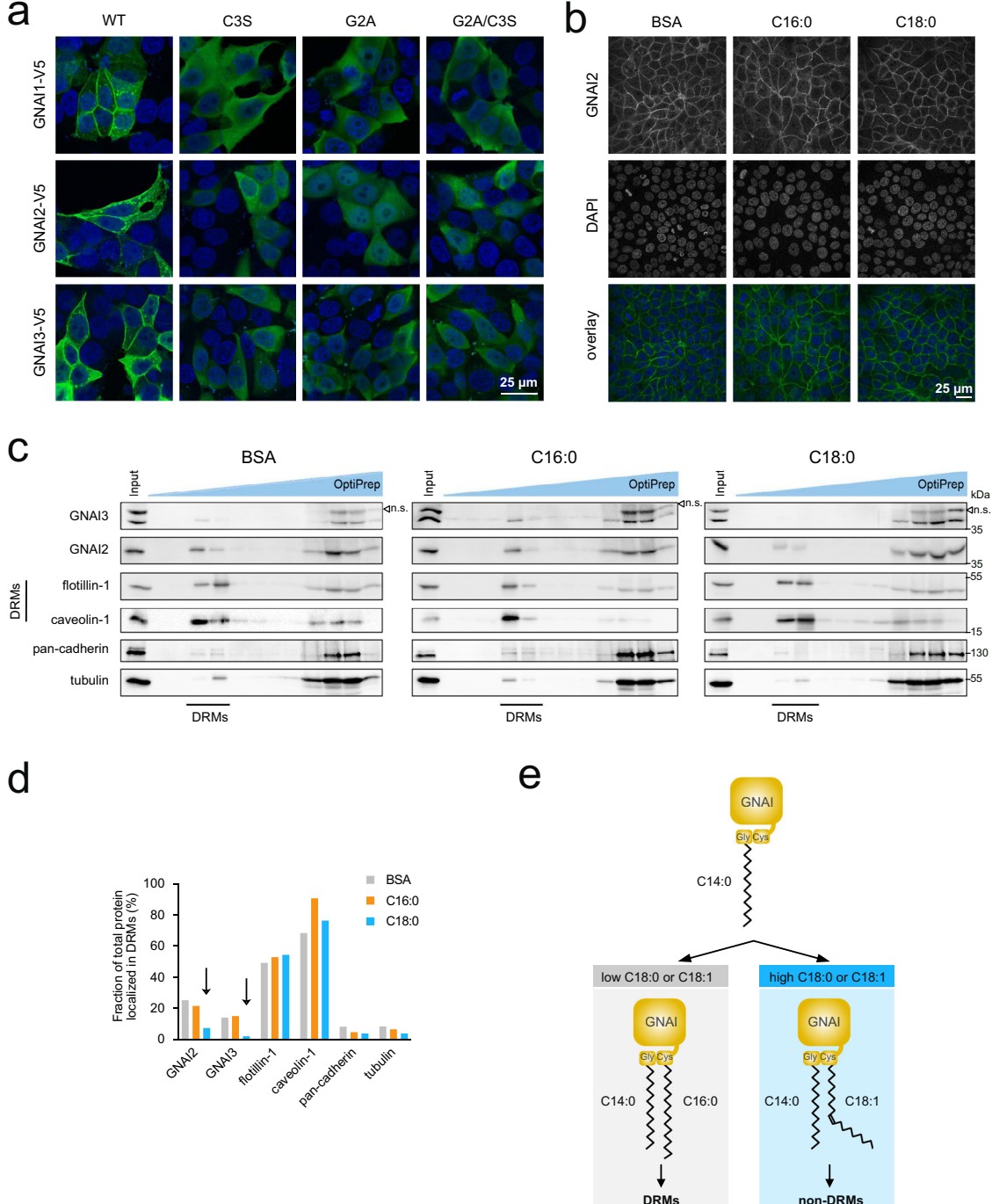

**Fig. 4 Exposure of cells to C18:0 shifts GNAI proteins out of detergent-resistant membranes (DRMs). a** S-acylation of GNAI proteins is required for their membrane localization. Unlike wild-type (WT) GNAI proteins, GNAI mutants lacking S-acylation do not localize to the plasma membrane, detected by immunostaining (green: anti-V5, blue: DAPI, scale bar 25 μm). Representative of three biological replicates. **b** Exposure of cells to C18:0 or C16:0 does not alter the localization of endogenous GNAI2 to the cell membrane, detected by immunostaining (green: anti-GNAI2, blue: DAPI, scale bar 25 μm). Representative of two biological replicates. **c**, **d** Endogenous GNAI2 and GNAI3 are depleted from DRMs upon exposure of cells to C18:0 but not C16:0. DRMs are isolated by the protein flotation assay in discontinuous OptiPrep gradients after solubilization with 1% Triton X-100 (**c**). As markers of DRMs, flotillin-1 and caveolin-1 were used. Quantification of the relative amount of each protein in DRMs normalized to total cell content of that protein (sum of all fractions) is shown in (**d**). Representative of three biological replicates. **e** Schematic diagram summarizing the findings presented here on GNAI S-acylation.

previously reported observations. C18:0 has been demonstrated to have anti-proliferative, anti-tumor and anti-metastatic effects both in vitro and in vivo in several animal models of spontaneous and chemically-induced tumorigenesis[16–25,69–71]. The AKT pathway integrates pro-survival and mitogenic stimuli to promote cellular growth and proliferation and PI3K/AKT hyperactivation

is frequently found in human tumors[68]. In gliomas, over-expression of *GNAI1* or *GNAI3* appears essential for AKT hyperactivation and tumor growth[32,40]. Hence if the effect of C18:0 on AKT activation that we see in cell culture carries over to the organism, this could explain in part the anti-tumor effects of C18:0. We recently showed that the effects of C18:0 on

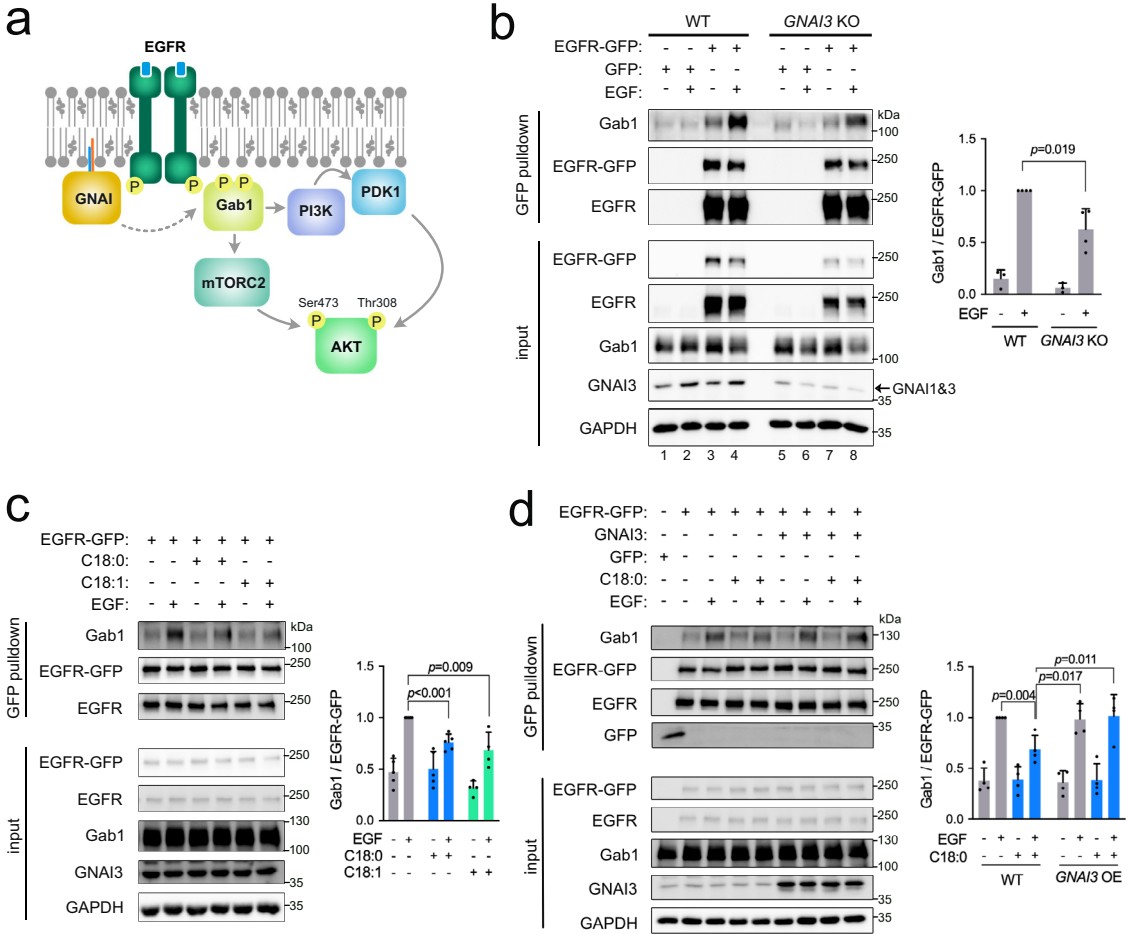

**Fig. 5 C18:0 and C18:1 reduce recruitment of Gab1 to EGFR via GNAI proteins. a** Schematic diagram of the role of GNAI in EGFR signaling. **b** The interaction of EGFR-GFP with Gab1 induced by EGF treatment (50 ng/mL for 5 min) is reduced in *GNAI3* KO MCF7 cells. Quantified is the amount of Gab1 in the pulldown normalized to the amount of EGFR-GFP in the pulldown. Data are presented as mean ± SEM of four biological replicates, *p*-values determined by two-sided *t*-test with correction for multiple comparisons using the Holm–Sidak method. **c** Exposure of MCF7 cells to C18:0 or C18:1 (both 100 μM, 3 h) blunts the interaction of EGFR-GFP with Gab1 induced by EGF (50 ng/mL, 5 min). Quantified is the amount of Gab1 in the pulldown normalized to the amount of EGFR-GFP in the pulldown. Data are presented as mean ± SEM of four biological replicates, *p*-values determined by two-sided *t*-test with correction for multiple comparisons using the Holm–Sidak method. **d** Overexpression of *GNAI3* rescues the effect of C18:0 (100 μM, 3 h) on Gab1 recruitment to EGFR-GFP upon EGF stimulation (50 ng/mL for 5 min). Quantified is the amount of Gab1 in the pulldown normalized to the amount of EGFR-GFP in the pulldown. Data are presented as mean ± SEM of four biological replicates and the *p*-values were calculated by two-sided *t*-test with correction for multiple comparisons using the Holm–Sidak method.

mitochondrial morphology, which we originally observed in cell culture[14], can also be observed in vivo in human subjects[15], indicating that cells in vivo in humans are poised to respond to changes in dietary C18:0 levels. Likewise, it was observed 20 years ago that stearate inhibits cell proliferation in response to EGF stimulation, via an unknown mechanism that does not involve cAMP[23]. Our data suggest this may happen via S-oleoylation of GNAI proteins. Likewise, oleic acid has been shown to reduce EGF signaling[72], and our data may provide one molecular mechanism for this effect.

## Methods
**Cell culture and treatments**. MCF7 cells, purchased from ATCC (ATCC HTB-22), were cultivated in high-glucose DMEM (Gibco) supplemented with 10% FCS (Sigma-Aldrich), 1% non-essential amino acids (Gibco), 100 U/mL Penicillin and 100 μg/mL Streptomycin (Gibco) in a 37 °C incubator under stable 5% $CO_2$ atmosphere. For siRNA transfection, Lipofectamine RNAiMAX transfection reagent (Thermo Scientific) was mixed according to the manufacturer's instructions with siRNAs (Dharmacon; Supplementary Data 1) in Opti-MEM reduced serum medium (Gibco) and cells were used in an experiment after 48 h. For transfection of plasmids, Lipofectamine 2000 transfection reagent (Thermo Scientific) was used 24 h prior to an experiment. Single clones stably expressing a

protein from pcDNA3-based vectors were isolated and maintained upon addition of geneticin (0.5 mg/mL, Gibco). In experiments where transient transfection was combined with fatty acid treatment, Lipofectamine RNAiMAX and 2000 were replaced by lipid-free transfection reagents Viromer Blue and Red (Lipocalyx), respectively. For use in cell culture, fatty acids and their derivatives were conjugated to BSA[14]. Briefly, fatty acids were dissolved to a concentration of 50 mM in 100 mM NaOH at 95 °C. Once in solution, 100 μL of fatty acid were dropwise added onto 600 μL of 10% fatty acid-free BSA pre-warmed to 50 °C. After mixing, the volume was filled up to 1 mL with water to reach the final concentration of fatty acid of 5 mM. BSA-conjugated fatty acids were applied on cells at a final concentration of 100 μM. Original and derived cell lines were tested negative for mycoplasma contamination (Eurofins Genomics, Germany). Cell lines were authenticated using Multiplex Cell Authentication by Multiplexion (Heidelberg, Germany) as described recently[73].

**Cloning and site-directed mutagenesis of GNAI1/2/3 and ZDHHC7**. Sequences of primers used in this study are provided in Supplementary Data 2. The open reading frames of human *GNAI1*, *GNAI2*, *GNAI3*, and *ZDHHC7* were PCR amplified from cDNA of MCF7 cells with primers containing EcoRI and BamHI (GNAI1-3) or NotI (ZDHHC7) sites using Phusion High-Fidelity Polymerase (Thermo Scientific), and cloned into a pcDNA3 backbone containing a C-terminal (*GNAI1-3*) or N-terminal (*ZDHHC7*) V5-tag. Due to the presence of an internal BamHI restriction site in the *GNAI2* sequence, bacterial clones with the correct construct were pre-selected by colony PCR in this case. To mutate the N-termini of

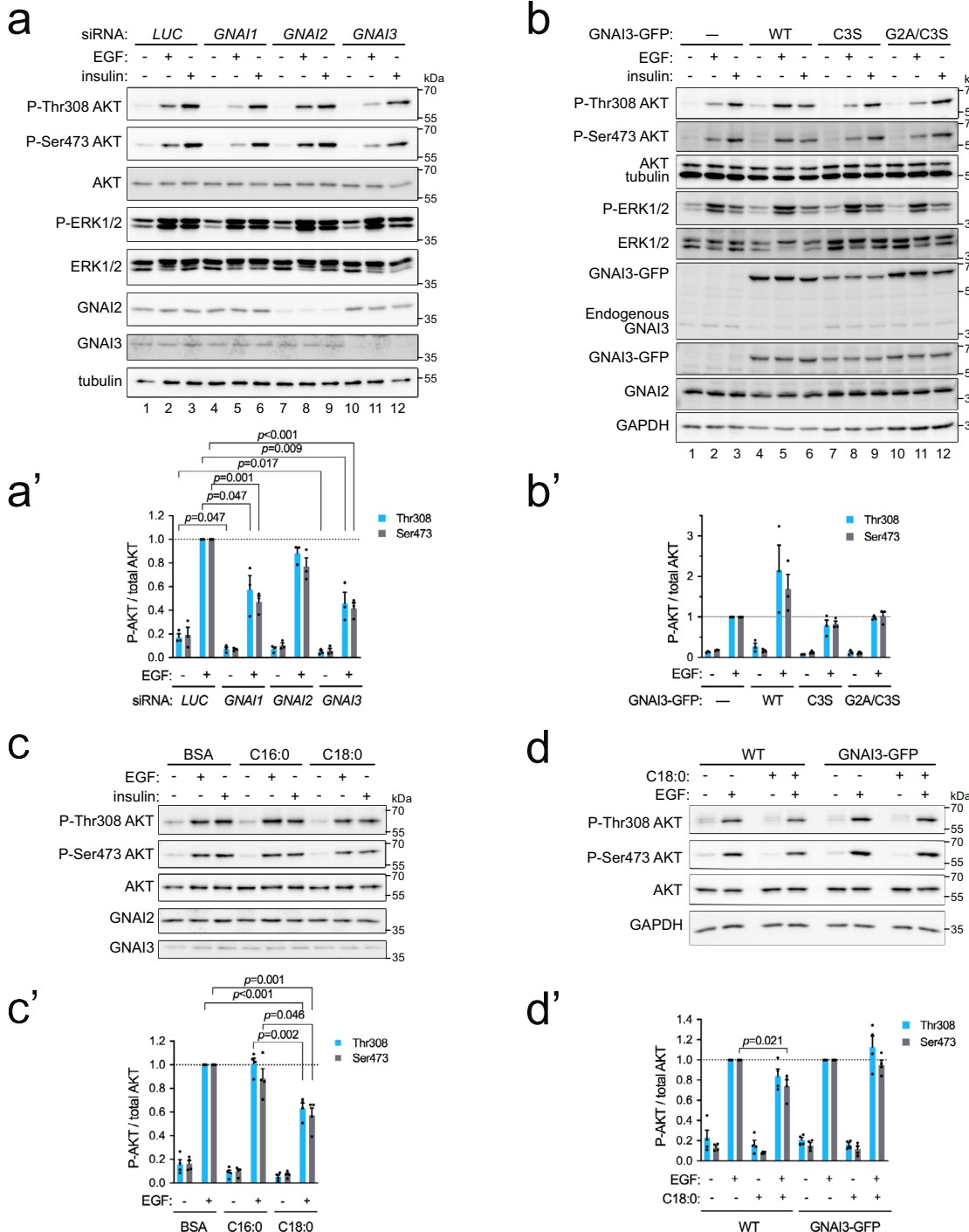

**Fig. 6 Exposure of cells to C18:0 reduces AKT activation in response to EGF. a–a'** GNAI1 and GNAI3 are required for maximal AKT activation in response to EGF. Two days after siRNA transfection, cells were treated with 50 ng/mL EGF or 300 ng/mL insulin for 10 min at 37 °C. Quantified in (**a'**): Data represent mean ± SEM of three biological replicates, *p*-values were determined by two-sided *t*-test with correction for multiple comparisons using the Holm–Sidak method. **b–b'** GNAI3 needs to be S-acylated to potentiate EGFR signaling. Overexpression of wild-type *GNAI3-GFP* but not mutants lacking S-acylation enhances AKT activation in response to EGF. As a control, the parental cell line (—) from which the stable GNAI3-GFP expressing cell lines were derived, is used. Quantified in (**b'**): mean ± SEM of three biological replicates. **c–c'** Exposure of cells to stearic acid suppresses EGF-induced activation of AKT. Cells treated 24 h with 100 μM BSA-conjugated stearate (C18:0) or palmitate (C16:0). Quantified in (**c'**): mean ± SEM of four biological replicates, *p*-values determined by two-sided *t*-test with correction for multiple comparisons using the Holm–Sidak method. **d–d'** Stable overexpression of *GNAI3-GFP* rescues the effect of C18:0 on AKT activation. Cells were incubated with C18:0 (100 μM, 24 h) and then treated with 50 ng/mL EGF for 10 min. As a control, the parental cell line (WT) from which the stable GNAI3-GFP expressing cell line was derived, is used. Quantified in (**d'**): mean ± SEM of four biological replicates, *p*-values determined by two-sided *t*-test with correction for multiple comparisons using the Holm–Sidak method.

*GNAI1/2/3*, primers with the desired mutation were used to PCR amplify *GNAI1/2/3* from pcDNA3-*GNAI1/2/3-V5* and the amplified mutated inserts were cloned into the same pcDNA3 backbone containing a C-terminal V5-tag. For *GNAI1* and *GNAI3*, EcoRI and BamHI restriction sites were used again, which covers the whole open reading frame. In the case of *GNAI2*, EcoRI and NheI restriction sites were used to insert a mutated N-terminal part of *GNAI2* into pcDNA3-*GNAI2-V5*. The *GNAI3* wild type and mutant inserts were PCR amplified from pcDNA3-*GNAI3-V5* plasmids using primers with EcoRI and BamHI restriction sites and subcloned into a pcDNA3 backbone containing a C-terminal GFP-tag. To mutate the active site of *ZDHHC7*, primers with the desired mutation (C160A) were used to PCR amplify the whole *V5-ZDHHC7* containing plasmid. The sequence of all constructs was confirmed by sequencing. Oligos were designed using Primer-BLAST (https://www.ncbi.nlm.nih.gov/tools/primer-blast) and sequence manipulation was performed using A plasmid Editor (ApE) software.

**Generation of GNAI3 KO cells using CRISPR technology.** A CRISPR design tool (http://crispr.mit.edu) was used to design sgRNAs targeting the first exon of human *GNAI3*[74]. The specificity of designed sgRNAs was checked with Cas-OFFinder by CRISPR RGEN Tools (http://www.rgenome.net/cas-offinder/)[75]. Oligonucleotides were synthesized and subcloned into the PX459 vector using BbsI restriction sites. The sequences of final plasmids were verified by sequencing using LKO.1 5′ primer. To transfect the plasmids into MCF7 cells, Lipofectamine 2000 (Thermo Fisher) was used according to the manufacturer's instructions. Following single-cell dilutions, isolated single clones were screened by immunoblotting for the absence of GNAI3 protein. Genomic DNA was isolated from the suspected knockout clones and the region targeted by sgRNA was amplified by PCR. Using the TOPO TA Cloning Kit (Thermo Fisher), PCR products were inserted into pCRII-TOPO vector and sequenced with the M13 forward primer in a number sufficient to assess the genotype of all three GNAI3 alleles. In the selected clone, three different frameshift mutations resulting in an early stop codon were detected.

**Protein electrophoresis and western blotting.** Cells were lysed in RIPA buffer (150 mM NaCl, 1% (v/v) Nonidet P-40, 1% (w/v) sodium deoxycholate, 0.1% SDS, 50 mM Tris-Cl pH 8.0) supplemented with a protease inhibitor cocktail (Roche) and 100 U/mL benzonase (Merck), and if phosphorylated proteins were to be detected, also with the PhosSTOP phosphatase inhibitor cocktail (Roche), 100 mM sodium fluoride, 2 mM sodium orthovanadate, and 64 mM beta-glycerophosphate. Protein concentrations were determined by BCA assay (Pierce BCA Protein Assay Kit, Thermo Fisher Scientific) using BSA as a standard. Lysates were denatured by incubation with Laemmli sample buffer for 5 min at 95 °C and separated with Tris-glycine SDS-PAGE. Proteins were transferred onto a nitrocellulose membrane and detected with primary antibodies according to manufacturers' instructions followed by incubation with HRP-conjugated secondary antibodies diluted 1:10,000. Chemiluminescence was recorded with the Chemidoc Imager (Bio-Rad) and quantified using Image Lab software (Bio-Rad).

Rabbit anti-AKT (#9272, 1:1000), rabbit monoclonal anti-caveolin-1 (#3267, 1:1000), rabbit monoclonal anti-CD71 (Transferrin receptor 1; #13113, 1:1000), rabbit monoclonal anti-c-Myc/N-Myc (#13987, 1:1000), rabbit monoclonal anti-EGFR (#4267, 1:1000), rabbit monoclonal anti-flotillin-1 (#18634, 1:1000), rabbit anti-Gab1 (#3232, 1:1000), rabbit monoclonal anti-GAPDH (#2118, 1:1000), mouse monoclonal anti-p44/p42 MAPK (Erk1/2; #9107, 1:1000), rabbit monoclonal anti-pan-cadherin (#9107, 1:1000), rabbit anti-phospho-AKT (Ser473; #9271, 1:1000), rabbit monoclonal anti-phospho-AKT (Thr308; #2965, 1:1000), rabbit monoclonal anti-phospho-p44/p42 MAPK (Thr202/Tyr204; #4370, 1:1000) antibodies were purchased from Cell Signaling Technology. Rabbit monoclonal anti-GNA1 (#ab140125, 1:1000), rabbit monoclonal anti-GNAI2 (#ab137050, 1:1000), rabbit monoclonal anti-GNAI3 (#ab173527, 1:1000), and mouse monoclonal anti-PDI (#190883, 1:1000) antibodies were purchased from Abcam. Mouse monoclonal anti-GM130 (#610823, 1:1000) antibody was purchased from BD Biosciences, mouse monoclonal anti-V5 tag (#R960-25, 1:1000) from Thermo Fisher, and mouse monoclonal anti-α-tubulin (#T9026, 1:3000) from Sigma-Aldrich.

**Acyl-PEG exchange assay.** To detect S-acylation, the acyl-PEG exchange assay, a method based on selective maleimide-labeling of acylated cysteines causing a mass shift, was performed as described previously[46,76], using methoxypolyethylene glycol maleimide (mPEG, 5 kDa).

**Detection of lipid modifications based on click chemistry.** C15:0- and C17:0-azide modified proteins were pulled down using the Click Chemistry Capture Kit (Jena Bioscience) and alkyne agarose beads (Jena Bioscience). Cells were seeded in 10 cm culture dishes. For Fig. 1d, upon reaching 80% confluence, cells were incubated with DMEM supplemented with delipidated FBS[14] for 24 h to starve them for fatty acids. For all other experiments which were done subsequently, this starvation step was skipped because it is not necessary. Afterward, cells were incubated for 3 h with 100 μM fatty acid analogs conjugated to BSA. The following fatty acid azides were used: C15:0-azide (15-azidopentadecanoic acid) from Life Technologies and C17:0-azide (17-azidoheptadecanoic acid) from ref. [14]. Treatment with BSA was used as a negative control. After treatment, cells were briefly

rinsed with PBS equilibrated to room temperature and lysed in 1 mL of urea-based lysis buffer supplemented with a protease inhibitor cocktail (Roche). Lysates were transferred into tubes, sonicated on ice for 30 s (10% amplitude) and cleared by centrifugation at $20,800 \times g$ for 5 min at 4 °C. The click reactions were assembled according to the protocol provided by the manufacturer, i.e., 800 μL lysate was mixed with 200 μL alkyne agarose beads and 1 mL reaction mixture. Samples were rotated end-over-end for 16–20 h at room temperature. Agarose beads were washed with $H_2O$, pelleted at $1000 \times g$ for 1 min and transferred into columns for extensive washing with the agarose washing buffer. Washed beads were resuspended in $H_2O$ and transferred into tubes again. Beads were pelleted at $1000 \times g$ for 1 min and resuspended in 2× Laemmli sample buffer supplemented with 1 M hydroxylamine to elute bound proteins. Samples were incubated for 15 min at room temperature and then boiled at 95 °C for 5 min. Beads were pelleted at $1000 \times g$ and the eluate was transferred into new tubes. Pulled-down proteins were loaded on a gel together with cell lysates that were kept as inputs. Western blots were probed with antibodies to test for the presence of specific proteins in pull-downs.

**Detection of lipid modifications by MALDI-TOF mass spectrometry.** The nature of fatty acyl residue on purified GNAI3 was analyzed by MS. To purify large amounts of GNAI3, we employed GFP-binding protein (GBP)[77] that can pull down GNAI3-GFP with high affinity and selectivity. Briefly, GBP was expressed in BL21 competent bacteria (Rosetta) from pET28c plasmid carrying 6xHis-GBP, purified using Ni-NTA agarose beads, dialyzed, and covalently coupled to activated sepharose beads according to the manufacturer's instructions. MCF7 cells with stable expression of GNAI3-GFP were harvested by trypsinization and lysed in lysis buffer (20 mM Tris-Cl, 800 mM NaCl, 0.5 mM EDTA, 1% NP-40, pH 7.5) supplemented with a protease inhibitor cocktail (Roche) and palmostatin B, a deacylase inhibitor, to reach a final concentration of 250 μM (EMD Millipore Calbiochem InSolution APT1 inhibitor palmostatin B). Cleared lysates were diluted 1:2 with a dilution buffer (20 mM Tris-Cl, 150 mM NaCl, 0.5 mM EDTA, pH 7.5) and mixed with GBP-sepharose beads. Purification was performed for 2 h at 4 °C with end-over-end rotation. Sepharose beads were washed 4 times with wash buffer (20 mM Tris-Cl, 300 mM NaCl, 0.5 mM EDTA, pH 7.5) and purified proteins were eluted in non-reducing Laemmli buffer for 5 min at 95 °C. Eluted proteins were alkylated with 30 mM iodoacetamide (IAA) for 30 min in the dark, separated via SDS-PAGE, and transferred to a PVDF membrane followed by Ponceau S staining. The band corresponding to GNAI3-GFP was processed as described below (similar to ref. [78] with small modifications): It was cut into small pieces, rinsed quickly with 100–200 μL of 40 % acetonitrile/50 mM $NH_4HCO_3$ (pH 7.5), and digested with 10 μL of trypsin (Promega, 15 μg/μL in 80 % acetonitrile/20 mM $NH_4HCO_3$, pH 7.5) for 1 h at 37 °C. The water-acetonitrile solution containing hydrophilic and slightly hydrophobic peptides was transferred to a clean tube and an aliquot (1–2 μL) was taken to confirm the protein identity by MALDI-TOF MS analysis and following Mascot server (www.matrixscience.com) search. The main part was frozen until combining with organic eluates and until further analysis. To release strongly hydrophobic peptides, the membrane pieces were further covered with a mixture of 5–10 μL of hexafluoroisopropanol (HFIP) and 10 μL of chloroform for 2 h to overnight. The water-acetonitrile and HFIP/chloroform extracts were combined, 15–20 μL of 0.5% trifluoroacetic acid/water solution was added to get phase separation and remove excess of buffer salts, and the organic phase enriched with various hydrophobic peptides was collected for MALDI-TOF MS analysis. Sample aliquots (1 μL) were mixed on a steel target with an equal volume of 2,5-dihydroxybenzoic acid in 30 % acetonitrile/0.5 % trifluoroacetic acid water solution (10 mg/mL). Mass spectra were recorded with a Ultraflextreme MALDI TOF-TOF mass spectrometer (Bruker Daltonik, Germany) equipped with a 355 nm (Nd) laser. The MH$^+$ molecular ions were measured in reflector mode to determine monoisotopic peak masses with the accuracy within 30 ppm. Fragmentation spectra were obtained in LIFT mode, and the accuracy of daughter ions measurement was within 1 Da. Mass spectra were processed using FlexAnalysis 3.2 software (Bruker Daltonik, Germany) and analyzed manually.

**GFP affinity co-purification.** To detect proteins interacting with EGFR-GFP, GFP affinity purifications using GFP-binding protein (GBP) coupled to sepharose beads were employed, based on ref. [77]. The protocol was modified in comparison to GNAI3-GFP purification for MS analysis (see above). Wild type or GNAI3 KO MCF7 were transiently transfected with a plasmid carrying the EGFR-GFP construct or GFP only as a control using Viromer Red according to the manufacturer's instructions. The next day, cells were treated with BSA-conjugated C18:0 or C18:1 (100 μM, for 3 h or 24 h as indicated). Afterward, the culture medium was exchanged for PBS and cells were treated with 50 ng/mL EGF for 5 min at room temperature. Then dithiobis(succinimidyl propionate) (DSP), a crosslinker, was added to reach the final concentration of 0.5 mM. After 30 min incubation at room temperature, the crosslinking reaction was quenched with 20 mM Tris-Cl pH 7.5 for 15 min. Cells were washed with PBS twice and lysed in an ice-cold lysis buffer (20 mM Tris-Cl, 800 mM NaCl, 0.5 mM EDTA, 1 % NP-40, pH 7.5) supplemented with protease inhibitor cocktail (Roche). Cleared lysates were adjusted with a dilution buffer (20 mM Tris-Cl, 150 mM NaCl, 0.5 mM EDTA, pH 7.5) to the same concentration and mixed with GBP-sepharose beads. Purification was performed overnight at 4 °C with end-over-end rotation. Next, the beads were washed five

times with wash buffer (20 mM Tris-Cl, 300 mM NaCl, 0.5 mM EDTA, 0.1% NP-40, pH 7.5) and purified proteins were eluted in 2x Laemmli buffer for 5 min at 95 °C. The presence of proteins of interest was tested by immunoblotting.

**Immunofluorescence.** Cells were seeded on cover slips in 24-well plates (50,000–80,000 cells per well) and transfected or directly processed the next day. Briefly, cells were fixed with 4% paraformaldehyde in PBS for 20 min, rinsed three times with ice-cold PBS, permeabilized with PBS + 0.2% (v/v) Triton X-100 for 10 min, and again rinsed three times with PBS. After blocking with 1% (w/v) BSA in PBS for 30 min, samples were incubated in a primary antibody diluted in 1% BSA in PBS according to the manufacturer's instructions for 1 h in a humidified chamber. Three 5 min washes with PBS were then followed by incubation in a secondary antibody diluted 1:1000 in 1% BSA in PBS for 1 h. After final three 5 min washes with PBS, cover slips were mounted on microscopic slides with Prolong Gold Antifade reagent with DAPI (Thermo Scientific). If not stated differently, all steps were carried out at room temperature. Representative images were recorded with Leica SP8 with ×63 objective and 2.35× digital zoom.

**Proliferation assay.** Cells were seeded in 96-well plates one day before starting an experiment to allow cells to attach to the plate (5000 cells per well). One plate was used for each day of the proliferation curve. In the plate for day 0, a standard curve of known cell number quantified with an automated cell counter (TC20, Bio-Rad) was prepared (0–40,000 cells per well). At a specific time of day, the media from the plate 0 were removed, cells were rinsed with PBS, PBS was removed and plates were stored at –80 °C. If necessary, a treatment was started in other plates at this point. A plate was collected every day at the same time as described above. At the end of the proliferation curve, DNA content was quantified using Hoechst 33258 (Cayman Chemical) as a proxy of cell number. Each well was incubated with 100 μL of H$_2$O for 1 h at 37 °C and afterward plates were placed at –80 °C until frozen. After thawing at 37 °C, a working dye solution (100 μL) was added to each well so that the final concentration was 1 μg/mL Hoechst 33258, 100 mM NaCl, 10 mM Tris-Cl, 10 mM EDTA (pH 7.4). Fluorescence was measured with the Tecan Infinite plate reader (excitation 350 nm, emission 455 nm). The blank value was subtracted and the proliferation curve was calculated using the standard curve from day 0.

**Detergent-resistant membranes.** Cells were harvested with trypsin, washed with PBS, and incubated for 30 min on ice in 1.2 mL of lysis buffer (0.25 M sucrose, 20 mM Tris-HCl pH 7.8, 1 mM magnesium chloride, 1 mM calcium chloride, 1% (v/v) Triton X-100) supplemented with a protease inhibitor cocktail (Roche). Detergent-resistant membranes were isolated from the postnuclear supernatant using discontinuous Opti-Prep gradients as described before[79,80]. After ultra-centrifugation at 100,000 × *g* for 16 h, gradients were divided into 1 mL fractions collected by Piston Gradient Fractionator. Each fraction was concentrated by TCA precipitation[81] and the protein precipitates were dissolved in 100 μL of 2x Laemmli loading buffer at 95 °C.

**RNA isolation and quantitative PCR.** RNA from cultured cells was isolated using TRIzol reagent (Invitrogen) according to the manufacturer's instructions. To produce cDNA, Maxima H Minus reverse transcriptase (Thermo Scientific) was used in combination with oligo(dT)$_{18}$ primers following the protocol provided by the manufacturer. Quantitative real-time PCR was performed using Maxima SYBR Green/ROX qPCR Master Mix (Thermo Scientific) and mRNA-specific primers listed in Supplementary Data 2. From the detected *Ct* values the respective $2^{-\Delta Ct}$ or $2^{-\Delta\Delta Ct}$ values were calculated. If not stated differently, the gene expression is normalized to *ACTB*.

**Sucrose gradients to separate membrane proteins.** Cells were harvested with trypsin, washed with PBS, and resuspended in 2 mL of lysis buffer (0.25 M sucrose, 1 mM magnesium sulfate, 10 mM Tris-HCl pH 7.5) supplemented with a protease inhibitor cocktail (Roche). The cell suspension was then homogenized with a cooled Dounce glas-glas homogenizer (10 strokes with a loose pestle followed by 30 strokes with a tight pestle). The homogenate was centrifuged for 5 min at 600 × *g* at 4 °C. The postnuclear supernatant was transferred to a new tube and the sediment was re-homogenized in 2 mL of lysis buffer (50 strokes with the tight pestle). After centrifugation, the supernatants were pooled; the volume was adjusted to 6 mL and loaded onto a sucrose cushion (6 mL of 75% (w/v) sucrose in 10 mM Tris-HCl pH 7.5). After 40 min centrifugation at 100,000 × *g* at 4 °C using the Beckman ultracentrifuge (the swing-out rotor SW 40 Ti), the membrane fraction was collected from the interface of sucrose cushion and cell homogenate, brought to 50% sucrose in 2 mL and loaded at the bottom of a discontinuous sucrose gradient (from bottom to top: 1.35 mL of 45% sucrose, 1.8 mL of 40% sucrose, 1.8 mL of 35% sucrose, 1.8 mL of 30% sucrose, 1.8 mL of 25% sucrose in 10 mM Tris-HCl pH 7.5). The gradients were ultracentrifuged for 4 h at 150,000 × *g* at 4 °C using the Beckman ultracentrifuge (the swing-out rotor SW 40 Ti). After ultracentrifugation, 1 mL fractions were collected by the Piston Gradient Fractionator. Each fraction

was concentrated by TCA precipitation[81] and the protein precipitates were dissolved in 100 μL of 2x Laemmli loading buffer at 95 °C.

**Acyl-CoA determination in cells by LC-MS/MS.** Acyl-CoA species (Avanti Polar Lipids) used as standards or for calibrations were diluted to 5 μM in methanol/water 1:1 (v:v) containing 0.1% ammonium hydroxide. LC-MS grade solvents were used for extractions and MS analyses. Pentadecanoyl coenzyme A (C15:0-CoA) and 10Z-heptadecenoyl coenzyme A (C17:1(n7)-CoA) were used as internal standards. In addition, a calibration mix of palmitoyl coenzyme A (C16:0-CoA), stearoyl coenzyme A (C18:0-CoA), and 9Z-octadecanoyl coenzyme A (C18:1(n9)-CoA) standards was used to generate calibration curves. Acyl-CoAs were extracted using a single-phase extraction as described[82]. Lipid extractions were performed at 4 °C in Eppendorf safelock test tubes. 100 pmol of each internal standard was added either to cells or the calibration mix. Cells were grown in 10 cm culture dishes to ~90% confluency and incubated with BSA-conjugated fatty acids (final concentration 100 μM) for 3 h. After the treatment, cells on each plate were quickly rinsed with PBS and scraped into 500 μL of methanol/ammonium hydrogen carbonate 155 mM 1:1 (v:v). The lysate was then transferred into a test tube and snap-frozen in liquid nitrogen. Individual plates/samples were processed completely from rinsing to freezing one after each other. Up to 200 μL of each lysate was used for extractions, adding methanol/water 1:1 (v:v) containing 0.1% ammonia to a final volume of 200 μL. Sample amounts were adjusted based on phosphatidylcholine amounts (see below). Typically, ~200 μL of a 270 μM solution of PC was subjected to extractions for acyl-CoA. For standard titrations, 0, 12.5, 25, 50, 75, and 100 pmol of the calibration mix was transferred into separate test tubes using Hamilton syringes. Four hundred microliters of 155 mM ammonium hydrogen carbonate solution was added to either sample or standard test tubes, followed by 300 μL of acetonitrile and 100 μL of isopropanol. Samples were vortexed, sonicated in an ice bath, and incubated at 950 rpm on an Eppendorf Thermo mixer C for each 30 s at 4 °C. This procedure was performed three times. Afterward, the test tubes were centrifuged at 20,800 × *g* at 4 °C for 10 min in a benchtop centrifuge. The supernatants were transferred to 2 mL Eppendorf test tubes. The remaining pellets were subjected to one re-extraction. Pooled supernatants were evaporated in a Centrivap system with reduced pressure at 30 °C. The solid residues were reconstituted in 500 μL of methanol/water 1:1 (v:v) containing 0.1% ammonium hydroxide, sonicated, and vortexed briefly on ice. The solution was centrifuged at 20,800 × *g* at 4 °C for 10 min. The supernatants were transferred to new 1.5 mL Eppendorf test tubes. 100 μL of this solution was transferred to silanized glass inlets in glass vials and subjected to the autosampler (SIL-30AC) of a Nexera X2 UHPLC system (Shimadzu), coupled to a QTRAP 6500 + (Sciex).

Chromatographic separation was performed as described[82] using a Waters ACQUITY UHPLC BEH C8 column (130 Å, 1.7 μm, 2.1 × 100 mm) including a 5 mm pre-column of similar chemistry. The column oven temperature was set to 35 °C and 5 μL was injected. The gradient for separation was set to A (water, containing 15 mM ammonium hydroxide) and B (acetonitrile, containing 15 mM ammonium hydroxide) with a flow rate of 0.4 mL/min. The gradient was started with 5% B for 1 min, followed by an increase to 37% from 1 to 5 min, to 60% from 5 to 11 min. Within 0.5 min the gradient was set to 100% B. From 14.5 to 15 min the gradient was set back to starting conditions of 5% B, followed by a 3 min equilibration time until 18 min. For MS parameters and MRM transitions see Supplementary Data 3. The peak areas were assessed using Analyst 1.6.3 (Sciex). Acyl-CoA concentrations were calculated based on the intensity of the internal standard C17:1(n7)-CoA, and the linear regression analysis of the calibration points. For data normalization, acyl-CoA concentrations were normalized to total PC as the bulk membrane lipid. For the quantification of total PC, cells were subjected to lipid extractions using an acidic liquid-liquid extraction[83] using 25 pmol PC. For the quantification of total PC, cells were subjected to lipid extractions using an acidic liquid-liquid extraction[83] using 25 pmol PC (PC 13:0/13:0, PC 14:0/14:0, PC 20:0/20:0, PC 21:0/21:0, each 25 pmol; Avanti Polar Lipids) as internal standard mix. The final CHCL$_3$ phase was evaporated under a gentle stream of nitrogen at 37 °C. The extracts were resuspended in 10 mM ammonium acetate in 60 μl methanol. Five microliters aliquots of the resuspended lipid extracts were diluted 1:4 in 10 mM ammonium acetate in methanol in 96-well plates (Eppendorf twin tec 96) prior to measurement. The measurement was performed on a QTRAP 5500 (Sciex), with chip-based (HD-D ESI Chip, Advion Biosciences) electrospray infusion and ionization via a Triversa Nanomate (Advion Biosciences). PC species were analyzed employing precursor ion scanning (PREC 184, selecting for the choline phosphate head group) as described[84]. The amount for endogenous molecular lipid species was calculated based on the intensities of the internal standards and the respective lipid intensities were extracted using LipidView (Sciex). Additional details including instrument parameters, raw data, and calibration curves are provided in Supplementary Data 1.

**Statistical analyses.** Statistical analyses were done using GraphPad Prism 8 software.

**Figure preparation.** Figures were prepared using GIMP (https://www.gimp.org/) and Affinity Designer (https://affinity.serif.com/en-gb/).

**Reporting summary**. Further information on research design is available in the Nature Research Reporting Summary linked to this article.

## Data availability

The mass spectrometry proteomics data have been deposited to the ProteomeXchange Consortium via the PRIDE[85] partner repository (https://www.ebi.ac.uk/pride/) with the dataset identifiers PXD021517 and PXD027064. All other data generated in this study are provided in the Supplementary Information and Source data files. Source data are provided with this paper.

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

## Acknowledgements

We thank Antoni Martija (DKFZ) for sharing reagents for constructing the EGFR-GFP construct. This project was funded by the European Research Council (ERC) under the European Union's Horizon 2020 research and innovation program (grant agreement No. 724286 - C18Signaling) to A.A.T., by the Russian Foundation for Basic Research (RFBR) project number 20-54-12007 to L.V.K., and by the Deutsche Forschungsgemeinschaft (DFG, German Research Foundation) project number 278001972 – TRR 186 and project number 112927078 – TRR 83 to B.B. The MALDI-TOF MS facility was made available in the framework of Moscow State University Development Program PNG 5.13.

## Author contributions

H.N., M.V.S., A.F.-C., T.S., and C.L. performed experiments. H.N., M.V.S., A.F.-C., T.S., C.L., A.K.M., B.B., L.V.K., and A.A.T. designed experiments, analyzed data, and wrote the manuscript.

## Funding

## Competing interests

The authors declare no competing interests.
