## [Peer Review File · Nature Communications]

REVIEWER COMMENTS

Reviewer #1 (Remarks to the Author):

The manuscript by Nuskova et al focuses on the fatty acylation status of the inhibitory G protein alpha subunits (herein referred to as GNAI). These proteins are known to be dually acylated with myristate at the N-terminal Gly and palmitate at the adjacent Cys. Here the authors show that Cys3 can be acylated with other fatty acids, notably C18:1 oleate, likely produced by desaturation of stearate (C18:0) in cells. Treatment of cells with C18:0 results in movement of GNAI3 out of detergent-insoluble membrane domains, reduction of Gab1 binding to activated EGFR and downstream signaling. The authors conclude that these effects are due to alternative acylation of GNAI with oleate and suggest that this is linked to antitumor properties of stearate.

General comments

1. A strength of the study is the use of Mass Spec to directly identify lipid modifications on GNAI proteins. It is clear that, in addition to palmitate, oleate is attached to GNAI in cells grown in unsupplemented media, and that GNAI oleoylation can be increased by 18:0 or 18:1 supplementation. The finding that an SCD1 inhibitor reduces the 18:1 signal implies that conversion of 18:0 to 18:1 occurs first, followed by attachment to GNAI.

2. This is not the first report of an S-acylated protein being modified with fatty acids other than 16:0, as there are numerous reports of this occurring for other proteins in the literature. Moreover, the authors state that the effects of acylating proteins with different fatty acids on functional outcomes is not known, but there are previous reports in the literature that show that modification with unsaturated fatty acids alters S-acylated protein signaling (eg *J Immunol* 2003 170:2932; *JBC* 276:30987).

3. A general weakness of the study is that the authors rely on treatment of cells with stearate to conclude that the signaling effects they observe are specifically due to oleoylation of GNAs. There is still a population of palmitoylated GNAI protein in cells treated with 18:0 (Fig 2C). In the GNAI KO cells, the reductions in GAB1 recruitment to EGFR shown in Fig 4 are small (40% reduction) and could be due to the presence of GNAs modified with palmitate. A method to follow GNAI proteins acylated with specific fatty acids would be preferable (see below).

Specific comments

4. The presence of a mixed population of GNAI proteins heterogeneously acylated with different fatty acids makes it difficult to conclude that only the oleoylated form is responsible for the observed effects. If the authors follow detergent-resistant domain localization and EGFR recruitment of GNAI specifically labeled with either 16:0 or 18:1, then they conclude that one or the other species is responsible for the phenotype.

5. The authors resort to using 18:0 supplementation for their signaling experiments in Figs 4 and 5, arguing that oleoylated GNAI formed by conversion to 18:1 is responsible for the observed changes. In Fig 4C, it is not clear whether the reduction in Gab1 recruitment to the EGFR in cells treated with 18:1 is statistically significant. Moreover, the authors suggest that the lack of effect of 18:1 after 24 hrs is because it is metabolized more rapidly, but this could be rectified by re-feeding the cells with 18:1 every 6-8 hrs. If treatment of 18:1 supplemented cells with the SCD1 inhibitor has an effect, this would support the authors' contention that the oleoylated form of GNAI is regulating signaling.

6. The experiments purporting to show a reduction in cell proliferation with 18:0 are not convincing, as there is minimal effect (Suppl Fig 6).

7. Fig 1:

Panel b: the signal for the GNAI3 band in the pulldown is very weak and almost undetectable.

Panel c: Why is the C15:0 azide signal higher than the corresponding LUC control?

Panel e: Why are the GNAI2 bands in the pulldown lower in the lanes with G2A and G2A/C3S in the C17:0 azide treated samples?

Reviewer #2 (Remarks to the Author):

While lipids have been widely linked to various cancers for a long time, how different lipid-derived metabolites exert their functions on cell signaling is still unclear at the molecular level. The authors here showed that the competition of palmitoylation with oleoylation on the same site within GNAI regulates EGFR signaling, providing an interesting mechanism where different fatty acids function quite uniquely to modulate cell signaling. This result also indicated there might be also other examples in which acylations also compete to fine-tune cellular signaling transduction, perhaps linking our diet choice to diseases, especially cancers here. Therefore, the results are important and intriguing, and should be of interest the field.

However, the current version of the manuscript still lacks thorough mechanistic insight and full characterization of the phenomena. For example, how palmitoylation and oleoylation on Cys3 is regulated? Whether biosynthesis of oleoyl-CoA vs. palmitoyl-CoA is contributing to the effects,

besides dietary uptake of the lipids. In addition, it requires additional evidence to show whether this kind of competition scenario for GNAI indeed is implicated in vivo or in a physiologically relevant model (besides just cell growth). Therefore, the reviewer could not recommend the publication of this manuscripts in Nat Commun at this stage.

Major points:

1. Given that ZDHHC7 predominately transfer oleoyl, but not palmitoyl to cys3, it is speculated that different transferases could modify this site. The author only tested 3 ZDHHCs (ZDHHC3, 6,7). While ZDHHC7 was identified to transfer oleoyl to cys3, whether there is other ZDHHCs exerting the same function is unclear. More thorough analysis of other ZDHHCs would be ideal to figure out how palmitoylation and oleoylation are regulated. Additional experiments using ZDHHC7 loss-of-function mutants, or KO ZDHHC7 should also be performed to analyze the effects on GNAI signaling. In addition, whether the deacylation rate would be affected by oleoylation?
2. The author claimed the level of C18:0 lipid exposed to cells effects acylation status of cys3 within GNAI. Do you think this modification alteration is directly controlled by the level of C18:0 or mediated by corresponding ZDHHCs? It is still unclear how this acylation change responds to the level of C18:0. Is this through the concentration of pal-CoA vs. oleoyl-CoA? The author should at least show that exogenous C18:0 could alter the intracellular fatty acyl-CoA composition.
3. In figure 4b, it is not reasonable to study GNAI mediated association of EGFR-Gab1 by comparing IP results from WT and GNAI3 KO cells when input of EGFR-GFP and EGFR was not even (i.e. lane 4 vs 8). The same issue also occurred in Figure 4d.
4. To confirm C18:0 affects EGFR signaling via GNAI proteins, the author knockout GNAI3 to test whether C18:0 would impact Gab1 recruitment (Figure 4d). In GNAI3 KO cells, however, the recruitment of this docking proteins is too low to respond to C18:0 treatment. Overexpression of ZDHHC7 or GNAI3 would provide a stronger basal level of Gab1 recruitment, thereby mirroring the effects of oleate treatment.
5. Palmitoylation of GNAI proteins facilitates its partition into DRMS. However, treatment of C16:0 to cells did not show increasing P-AKT level when co-treated with EGF. Even more, without EGF stimulation, C16:0 treatment also decreased the EGFR signaling (Figure 5C). One possible reason is that effects of C16:0 is miscellaneous. Therefore, the author should identify transferases for palmitoylation or oleoylation of GNAI at cys3. Then, overexpression of corresponding ZDHHCs to confirm acylation of this sites indeed regulates downstream EGFR signaling.
6. The author should show whether C18:0 promotes cell proliferation through GNAI oleoylation. Although Suppl. Figure 6a-b suggested GNAI 1 and 3 was associated with cell proliferation, more

solid evidence is required to confirm such cell growth is through GNAI oleoylation when treated with C18:0, at least in part. For example, will the sensitivity to C18:0 treatment alter in the context of GNAI KO or overexpression? Will exogenous GNAI restore the sensitivity in GNAI-depleted cells?

7. In vivo study is suggested to show C16:0 and C18:0 modification of GNAI3 influence competitively tumor growth. It is also of interest to link different diets wherein the ratio of C16:0 and C18:0 vary to tumor growth in vivo.

Minor issues:

1. The author tested silencing of several ZDHHC palmitoyltransferases on the acylation level of GNAI2 and stated that ZDHHC7 is the main enzyme responsible for modifying GNAI2 in page 10. However, in Figure 1c, silencing of ZDHHC7 didn't reduce the acylation level of GNAI2. Technical issue exists within the group of ZDHHC7 knockdown, since no C15:0-azide or C17:0-azide treatment has strong signal (lane 10), the experiment should be repeated. Furthermore, it would be of higher scientific value to screen all the ZDHHCs to find the strongest ZDHHC responsible for GNAs. Since the basal level of GNAs fatty acylation is low, it's would be interesting to overexpress ZDHHCs to look at fatty acylation of GNAs. At the same time, any preferences of ZDHHCs among GNAI1, GNAI2 and GNAI3?

2. In Figure 3a, a membrane marker should be included for staining. Since the author potentiates the effects of fatty acylation on GNAs membrane localization, it would be better to treatment the cells w/o fatty acids to look at the change of GNAI membrane localization by IF.

3. In Figure 3c, the input level of GNAI3 is higher in BSA group, it would be suspicious that C16:0 and C18:0 reduces the fractions of DRMs was due to the reduced total protein level.

4. In Figure 4b and 4c, GNAI3 KO reduced the total protein level of EGFR (input), the reduced pulldown of EGFR and Gab1 may due to the reduced total protein level. It would be interesting to overexpress GNAI3 to check the response to EGF treatment.

5. In Figure 5a, silencing of GNAI1 is not confirmed.

6. In Figure 5c, C18:0 treatment has no effects on EGF induced phosphorylation of AKT, the statement in page 17 is not right.

7. In page 13, no in vivo experiment was performed in the current study, the author should correct the statement.

Technical issues:

In supplemental Figure 1a, GNAI3 antibody (ab173527) is not working for detection of GNAI3 but GNAI1. GNAI1 antibody (ab14015) only detects GNAI1, marginally detects GNAI3. Thus, the description on page 9 should be revised. At the same time, no antibody is working for GNAI3, it would be hard to detect endogenous GNAI3 and confirm the silencing effects of siRNA. The author should seek another GNAI3 antibody.

In Figure 1b, the signal of GNAI2 and GNAI3 pulldown is quite weak, the author should improve the quality of the blot.

Reviewer #3 (Remarks to the Author):

The manuscript by Nuskova et al claimed that stearic or oleic acid blunts EGFR and AKT signaling caused by the oleoylation of a cystein residue on GNAI. They proposed a mechanism by which oleoylation of GNAI leads to the exclusion of GNAI with EGFR in the detergent resistant membrane (DRM), in contrast to palmitoylation or stearylation on the same residue. The study is interesting and likely to shed new light into a postranslational modification of a protein by different fatty acid may affect the biological activities. However, there are some major concerns regarding this study and their final conclusions. First of all, if the oleoylation of C3 on GNAI is responsible for the potential beneficial effects of anti-tumor properties, one would argue that oleic acid is better than stearic acid in this regard. Accordingly, there are more human data to link C18:1 to health benefits than stearic acid. As the authors are aware that there are more negative health effect for stearic acid in humans than the "anti-tumor effects". Secondly, it remains to be clearly defined whether C18:1 oleoylation on C3 occurs on the protein or only oleic acid can be used as a substrate. Although the author attempted to use a desaturase inhibitor in their experiments, it is still inconclusive to tease this out. An alternative would be to carefully monitor the fatty acid composition by GC-MS in the cell, either in free form or esterified to see if C18:0 has to be desaturated before it is acylated onto the protein. With that said, it is still unclear how the the cells use this mechanism in vivo. C18:0 or C16:0 is the predominant fatty acid in the cell membrane. Why they have to use the exogenous fatty acid for this purpose? In other words, it is hard to imagine that the benefits of C18:0 is only observed after dietary supplementation as in the study cited in ref.14 when our body have plenty of C18:0. Below are some minor concerns:

1. The concentrations of azido fatty acids in the click experiments were not shown.
2. If C18:1 modification has such a dramatic effect, one would expect similar effect on C16:1 modification at the same site. Similiar modifications is observed in Fig.2C.
3. Figure 2 is critical for the identification of these novel modifications. Please label m/z in the upper panel in Fig.2C. Zoom-in the region between 1800 to 2000 would really help reader to see the

modifications. In the text, the authors mentioned a broad peak in these spectra but they are not clear.

4. The fragmentation in Fig.2D did not directly reflect the nature of the modification. These modifications are indirectly derived from the parent ion and y ions. Did the myristoylation of the G occur endogenously? What is the broad peak at 1750 shown up in panel 2, 3, and 5 of Fig. 2D?

5. In the first graph on P11, "In contrast, palmitoylation of GNAI2 was retained upon knockdown of either ZDHHC3 or ZDHHC7.." ZDHHC3 or ZDHH6?

6. Where does oleoylation occur in the cells? Does it happen before the protein transported to the membrane or on the membrane?

Reviewer #1

The manuscript by Nuskova et al focuses on the fatty acylation status of the inhibitory G protein alpha subunits (herein referred to as GNAI). These proteins are known to be dually acylated with myristate at the N-terminal Gly and palmitate at the adjacent Cys. Here the authors show that Cys3 can be acylated with other fatty acids, notably C18:1 oleate, likely produced by desaturation of stearate (C18:0) in cells. Treatment of cells with C18:0 results in movement of GNAI3 out of detergent-insoluble membrane domains, reduction of Gab1 binding to activated EGFR and downstream signaling. The authors conclude that these effects are due to alternative acylation of GNAI with oleate and suggest that this is linked to antitumor properties of stearate.

General comments

1. A strength of the study is the use of Mass Spec to directly identify lipid modifications on GNAI proteins. It is clear that, in addition to palmitate, oleate is attached to GNAI in cells grown in unsupplemented media, and that GNAI oleoylation can be increased by 18:0 or 18:1 supplementation. The finding that an SCD1 inhibitor reduces the 18:1 signal implies that conversion of 18:0 to 18:1 occurs first, followed by attachment to GNAI.

We thank the reviewer for this positive assessment.

We have added a lot of new data to address the Reviewer's criticisms below. We have added new data to Fig. 5c, showing that C18:1 treatment significantly reduces Gab1 recruitment, and multiple new click-iT pulldown assays (Fig. 1b, 3a, 3b, 3c, 3d) with significantly improved signal/noise. We think these new data have significantly improved the manuscript.

2. This is not the first report of an S-acylated protein being modified with fatty acids other than 16:0, as there are numerous reports of this occurring for other proteins in the literature. Moreover, the authors state that the effects of acylating proteins with different fatty acids on functional outcomes is not known, but there are previous reports in the literature that show that modification with unsaturated fatty acids alters S-acylated protein signaling (eg J Immunol 2003 170:2932; JBC 276:30987).

(To make sure there are no misunderstandings, we would like to kindly point out that we did not claim that this is the first report of differential acylation of a protein. In fact, we cite several such studies in the Introduction.)

We thank the reviewer for pointing out Liang et al. (JBC 276:30987) because this is indeed the only other study we can find that shows a functional consequence of differential S-acylation with different lipid species on protein function. We

now cite it in the manuscript Introduction and Discussion, and point out that together with our study this indicates a more general mechanism whereby differential acylation of proteins with different lipid species can affect protein function. We have also toned down the statement in this regard in the Abstract and in the Introduction. (Please note that in J Immunol 2003 170:2932 cited by the reviewer palmitoylation is only compared to no acylation, but not to differential acylation with different lipid species.)

3. A general weakness of the study is that the authors rely on treatment of cells with stearate to conclude that the signaling effects they observe are specifically due to oleoylation of GNAIs. There is still a population of palmitoylated GNAI protein in cells treated with 18:0 (Fig 2C). In the GNAI KO cells, the reductions in GAB1 recruitment to EGFR shown in Fig 4 are small (40% reduction) and could be due to the presence of GNAIs modified with palmitate. A method to follow GNAI proteins acylated with specific fatty acids would be preferable (see below).

Please see below (to Specific Comment 4) for a detailed response to a method for following GNAI proteins acylated with different fatty acid species.

That said, we do not think the reduced GAB1 recruitment caused by C18:0 treatment can possibly be explained by the presence of GNAIs modified with palmitate. In the control condition, most of the GNAI protein is palmitoylated with comparatively little oleoylation (Fig. 2c and new Fig. 3e). When cells are treated with C18:0, this reduces the amount of GNAI palmitoylation and increases the amount of oleoylation (Fig. 2d and 3e). Hence the reduced GAB1 recruitment observed upon C18:0 treatment results from reduced GNAI palmitoylation and increased oleoylation.

Specific comments

4. The presence of a mixed population of GNAI proteins heterogeneously acylated with different fatty acids makes it difficult to conclude that only the oleoylated form is responsible for the observed effects. If the authors follow detergent-resistant domain localization and EGFR recruitment of GNAI specifically labeled with either 16:0 or 18:1, then they conclude that one or the other species is responsible for the phenotype.

We would like to clarify that we do not believe that one acylated specie is responsible for the phenotype. Instead, they are both responsible for the phenotype in that it is the comparison between the two species that explains what is happening. In the control condition, most of the GNAI protein is palmitoylated and comparatively little is oleoylated. When cells are treated with C18:0, this reduces the amount of GNAI palmitoylation and increases the amount of GNAI oleoylation. Hence upon C18:0

treatment, the reduced localization of GNAI proteins to detergent-resistant membranes (DRMs) and the reduced Gab1 recruitment result from reduced GNAI palmitoylation and increased oleoylation. From this we conclude that oleoylated GNAI must be less capable compared to palmitoylated GNAI to reside in DRMs and to recruit GAB1. We do not really see another possible interpretation? This also fits with a previous study where recombinant GNAI proteins were purified from *E. coli*, acylated chemically in vitro, and loaded onto sphingolipid- and cholesterol-rich liposomes to find that acylation with saturated fatty acids leads to partitioning into DRMs whereas acylation with unsaturated fatty acids leads to exclusion from DRMs (PMID 10636925). That said, as detailed below, we tried hard to follow the DRM localization of GNAI labeled with either C16:0 or C18:1 as suggested by the review, but this turned out to be technically very challenging:

We tried several approaches:

1. Our first idea was to isolate GNAI3-GFP from DRMs and then determine the acylation status of this DRM-fraction directly by MS. This approach, however, did not yield enough protein for MS. For MS analysis of acylation, we need to purify GNAI3-GFP protein from cells growing in 2 15-cm dishes. For the optiprep gradients, we can run at most lysate from one 10cm dish per gradient otherwise the separation is not good (i.e. 1/6th the needed material). Furthermore, only 20% of the GNAI protein is in the DRMs (i.e. 1/30th the needed material). In addition, we saw that the presence of optiprep in the samples significantly decreases the efficiency of the GFP pulldown. Altogether, this made this approach unfeasible.

2. We tried to specifically label proteins with azido fatty acids and then investigate where they go in optiprep gradients by pulling down labeled proteins from the various fractions and immunoblotting for GNAI2/3. Here too, we did not get enough signal. Although during the revisions we significantly improved the efficiency and hence signal from the click-iT pulldowns (new Fig.1b and Fig. 3), nonetheless, just as for the MS, we need more material for the click-iT pulldown + immunoblot than what we can get from DRM fractionations.

3. We thought we could test more generally the localization of all proteins modified with individual lipid azides in/out of DRMs. It is possible that, generally, proteins modified with saturated FAs prefer to be in DRMs whereas proteins modified with unsaturated FAs prefer to reside outside of DRMs. The procedure would be to do DRM fractionations on azido-FA treated cells, and then perform the click reaction on the fractions either to biotin (to later detect with HRP-conjugated anti-biotin antibody and chemiluminescence) or to fluorescent TAMRA (for direct in-gel detection using a fluorescent reader) to see where the modified proteins reside. (Note that for our standard

click-it pulldowns we click the proteins onto beads, elute, and then detect individual proteins by immunoblot. In this case here, we need to detect all proteins with a particular azido-FA, so we need to detect the FA. Hence, the biotin or TAMRA approach.) To test feasibility, we first checked if we can detect modification of proteins by azido fatty acids in whole cell lysates of cells incubated +/- C17:0-azide. Unfortunately, there was no specific signal above background for either the biotin-chemiluminescence approach or the direct TAMRA approach (please see Reviewer Figures 1-2 below). We tested two different Click-iT protocols (#1 using commercial buffers and #2 with home-made buffers to scale-up the reaction). In none of the cases did we see a specific signal in the +C17:0-azide compared to the -C17:0-azide control.

   Ponceau anti-biotin    Click-iT #1 Click-iT #2 Click-iT #1 Click-iT #2   BSA: ++++ ++++   C17:0-azide: ++++ ++++   biotin-alkyne: +++ +++      		Ponceau				anti-biotin					Click-iT #1		Click-iT #2		Click-iT #1		Click-iT #2		BSA:	+	+	+	+	+	+	+	+	C17:0-azide:	+	+	+	+	+	+	+	+	biotin-alkyne:		+	+	+		+	+	+	Reviewer Figure 1: Detection of azido-FA labeled proteins by clicking alkyne-biotin on the proteins and then detecting with HRP-conjugated anti-biotin antibody.
	Ponceau				anti-biotin																																									
	Click-iT #1		Click-iT #2		Click-iT #1		Click-iT #2																																							
BSA:	+	+	+	+	+	+	+	+																																						
C17:0-azide:	+	+	+	+	+	+	+	+																																						
biotin-alkyne:		+	+	+		+	+	+																																						
   TAMRA COOMASSIE    short exposure long exposure Click-iT #1 Click-iT #2    Click-iT #1 Click-iT #2 Click-iT #1 Click-iT #2 Click-iT #1 Click-iT #2 Click-iT #1 Click-iT #2   BSA: ++++ ++++   C17:0-azide: ++++ ++++       		TAMRA				COOMASSIE					short exposure		long exposure		Click-iT #1		Click-iT #2			Click-iT #1	Click-iT #2	Click-iT #1	Click-iT #2	Click-iT #1	Click-iT #2	Click-iT #1	Click-iT #2	BSA:	+	+	+	+	+	+	+	+	C17:0-azide:	+	+	+	+	+	+	+	+	Reviewer Figure 2: Detection of azido-FA labeled proteins by clicking alkyne-TAMRA on the proteins followed by in-gel fluorescent detection.
	TAMRA				COOMASSIE																																									
	short exposure		long exposure		Click-iT #1		Click-iT #2																																							
	Click-iT #1	Click-iT #2	Click-iT #1	Click-iT #2	Click-iT #1	Click-iT #2	Click-iT #1	Click-iT #2																																						
BSA:	+	+	+	+	+	+	+	+																																						
C17:0-azide:	+	+	+	+	+	+	+	+																																						

4. FITC-labeled cholera toxin can be used to aggregate and detect DRMs in living cells. So we tried to incubate cells with azido fatty acids and FITC-labeled cholera toxin, and then perform the click-iT reaction to TAMRA to investigate the colocalization of FITC (DRMs) and TAMRA. Unfortunately, cholera toxin doesn't work well in all cell lines and it seems that in MCF-7 cells, which we use in this study, the aggregation of DRMs is not efficient (please see figure below). Moreover, it is difficult to see any increase in TAMRA fluorescence in C15:0- or C17:0-azide treated cells above the background in unlabeled cells:

In sum, although we tried hard what the reviewer suggested, we were not able to detect the subcellular localization of GNAI proteins S-acylated with specific FA species. In our opinion, all the approaches that we tested are not sensitive enough because only a fraction of the total pool of proteins is modified by the exogenous azido fatty acids. Hence we get enough signal if we concentrate proteins from whole cell lysates using alkyne beads (as in Figures 1 and 3) but as soon as we combine this

with a fractionation method or a method that does not concentrate protein, we are below the level of detection.

That said, as discussed above, we believe the results presented in the manuscript regarding the consequences of shifting GNAI acylation from palmitoylation towards oleoylation are interpretable.

5. The authors resort to using 18:0 supplementation for their signaling experiments in Figs 4 and 5, arguing that oleoylated GNAI formed by conversion to 18:1 is responsible for the observed changes. In Fig 4C, it is not clear whether the reduction in Gab1 recruitment to the EGFR in cells treated with 18:1 is statistically significant. Moreover, the authors suggest that the lack of effect of 18:1 after 24 hrs is because it is metabolized more rapidly, but this could be rectified by re-feeding the cells with 18:1 every 6-8 hrs. If treatment of 18:1 supplemented cells with the SCD1 inhibitor has an effect, this would support the authors' contention that the oleoylated form of GNAI is regulating signaling.

The reviewer raises an important point that in the original manuscript the effect of treating cells with C18:1 on Gab1 recruitment was visible (and similar in magnitude to the C18:0 effect) but not statistically significant due to a slightly larger spread in the data. We have now repeated the experiment and the result is significant (now Fig 5c).

We believe this experiment, which now clearly shows that 3 hours after treating cells with C18:1 Gab1 recruitment is reduced, is better than a 24h treatment with re-feeding every 6-8 hours because conceptually it makes exactly the same point, except that in the 24h timeframe there is more possibility for secondary effects to take place (e.g. transcriptional changes translating into proteome changes, etc.)

(Regarding treating cells with C18:1 plus SCD1 inhibitor, it is not clear to us what the concept of this experiment would be, given that C18:1 is already desaturated ?)

6. The experiments purporting to show a reduction in cell proliferation with 18:0 are not convincing, as there is minimal effect (Suppl Fig 6).

We agree with the reviewer that the magnitude of the C18:0 effect is not very large. That said, the result is statistically significant ($p < 0.01$), and the C18:0 treatment causes a drop in cell count by almost 20%. This is in line with the magnitude of the effect we see on Gab1 recruitment and Akt phosphorylation. We believe it is important to keep in mind that we are looking

here at physiological regulation. If every time we ingest C18:0 this would shut off EGFR signaling more strongly in our cells, this would probably cause developmental defects in us and perhaps lethality. Hence the magnitude of this effect should not be expected to be as large as the consequences of pharmacological inhibition or gene knockdowns.

7. Fig 1:

Panel b: the signal for the GNAI3 band in the pulldown is very weak and almost undetectable.

We have optimized the click-iT pulldown of proteins modified with azido fatty acids so that we get more signal. We repeated the experiment and replaced Figure 1b with new Western blots that have a much better signal/noise ratio.

Panel c: Why is the C15:0 azide signal higher than the corresponding LUC control?

We are not sure what the reviewer means. The control for the C15:0 azide is the treatment with BSA only, not the LUC control. In panel C (now Figures 3a-d), the lanes were arranged in triplets. For instance, lane 1 is the BSA control, lane 2 is C15:0-azide and lane 3 is C17:0-azide. The signal in lane 2 was not higher than in lane 1. The only exception was upon ZDHHC7 knockdown where there was a bit more background in the BSA control compared to the C17:0 azide treatment, because the ZDHHC7 knockdown reduces GNAI modification with C17:0-azide.

As mentioned above, however, we improved the click-iT pulldown so we now have a better signal/noise ratio. We repeated and extended this experiment and the new Western blots are presented in Figures 3a-d. We hope this resolves the issue raised by the reviewer.

Panel e: Why are the GNAI2 bands in the pulldown lower in the lanes with G2A and G2A/C3S in the C17:0 azide treated samples?

(Now Figure 1d). This is indeed an interesting observation. We do not know why. It is possible that expression of the G2A-mutant forms are 'trapping' the transferase and thereby reducing modification of endogenous GNAI2. This, however, is speculative and would require further work to see if it is true.

Reviewer #2:

While lipids have been widely linked to various cancers for a long time, how different lipid-derived metabolites exert their functions on cell signaling is still unclear at the molecular level. The authors here showed that the competition of palmitoylation with oleoylation on the same site within GNAI regulates EGFR signaling, providing an interesting mechanism where different fatty acids function quite uniquely to modulate cell signaling. This result also indicated there might be also other examples in which acylations also compete to fine-tune cellular signaling transduction, perhaps linking our diet choice to diseases, especially cancers here. Therefore, the results are important and intriguing, and should be of interest the field.

We thank the reviewer for the positive evaluation.

However, the current version of the manuscript still lacks thorough mechanistic insight and full characterization of the phenomena. For example, how palmitoylation and oleoylation on Cys3 is regulated? Whether biosynthesis of oleoyl-CoA vs. palmitoyl-CoA is contributing to the effects, besides dietary uptake of the lipids. In addition, it requires additional evidence to show whether this kind of competition scenario for GNAI indeed is implicated in vivo or in a physiologically relevant model (besides just cell growth). Therefore, the reviewer could not recommend the publication of this manuscripts in Nat Commun at this stage.

We have now added to the manuscript several experiments to address these issues.

One general issue raised by the reviewer is how palmitoylation vs oleoylation of GNAI Cys3 is regulated. The acylation of GNAI3 is an example of a post-translational modification that results from the covalent attachment of a metabolite to a protein. In some cases, the stoichiometry of such PTMs depends on the activity of the enzymes that catalyze the modifications, whereas in other cases it depends on the abundance of the corresponding metabolites. An example of the first case is protein phosphorylation, which depends on the activity of kinases and not on cellular ATP concentration. In contrast, protein acetylation is strongly influenced by cellular levels of acetyl-CoA. Likewise, O-GlcNAcylation of proteins is also mainly regulated by O-GlcNAc concentrations. Both of these examples provide a mechanism how cellular metabolites directly regulate cell signaling. With this in mind, we now added a large number of new experiments to dissect whether GNAI S-acylation is regulated by the ZDHHC acyl-transferase enzymes or by the levels of the lipids. All the data we obtained indicate it is the lipid levels that matter:

1. In new Figure 3f we now show that exposure of cells to C16:0, C18:0 or C18:1 in the medium causes an increase in the corresponding pool of intracellular acyl-CoA, as suggested by this Reviewer in Major Point 2.

2. In new Figure 3e, we directly assayed the consequence of this on GNAI acylation by mass spectrometry on GNAI protein isolated from cells treated with control medium, C16:0-containing medium or C18:0-containing medium. This shows that exposure of cells to C16:0 or C18:0 shifts GNAI acylation towards palmitoylation or oleoylation respectively.

This is consistent with results from our original submission, where we provided mass spec. data showing that when C¹³-labeled C18:0 is added to cells, this causes GNAI acylation to shift from one predominant peak of C16:0-modified protein + one small C18:1-modified peak, to two equally sized peaks of C16:0 and ¹³C18:1 modified protein (now Fig. 2c-d). Hence total acylation on this site shifted towards oleoylation when cells were treated with ¹³C18:0.

3. In contrast, manipulation of the ZDHHC enzymes does not seem to modify the ratio of GNAI palmitoylation versus oleoylation. Knockdown of ZDHHC3 has no effect (new Fig. 3a) whereas knockdown of ZDHHC7 mildly reduces both GNAI palmitoylation and oleoylation, but the ratio does not seem to be altered (Fig. 3a). This is consistent with the fact that ZDHHC7 has previously been reported to use both C16:0 and C18:0 equally well as substrates (PMID 28167757). Double knockdown of ZDHHC3 and 7 does not decrease GNAI acylation further (new Fig. 3b), indicating that multiple ZDHHC enzymes are acting redundantly in vivo to acylate GNAI proteins.

4. Conversely, we overexpressed ZDHHC7 (as suggested by this reviewer in Point 4) and this also did not skew the acylation of GNAI2/3 towards palmitoylation or oleoylation (new Fig. 3c). Furthermore, it also did not increase total GNAI acylation, indicating that ZDHHC activity is not limiting for GNAI acylation in vivo.

5. We provide evidence in new Fig. 3c that ZDHHC7 binds both C15:0-azide and C17:0-azide on its active site cysteine, consistent with it able to use both lipids as substrates.

In sum, all the data we have indicate that the ratio of GNAI palmitoylation versus oleoylation is regulated by substrate availability and not enzyme activity.

This is consistent with previous work we did in which we showed via a clinical study that when people ingest C18:0, this activates a signaling pathway in vivo involving C18:0 attachment

to TfR1 and consequently mitochondrial fusion [PMID 30087348 and 26214738].

Major points:

1. Given that ZDHHC7 predominately transfer oleoyl, but not palmitoyl to cys3, it is speculated that different transferases could modify this site. The author only tested 3 ZDHHCs (ZDHHC3, 6, 7). While ZDHHC7 was identified to transfer oleoyl to cys3, whether there is other ZDHHCs exerting the same function is unclear. More thorough analysis of other ZDHHCs would be ideal to figure out how palmitoylation and oleoylation are regulated.

During the revision we significantly improved our click-it pulldowns so that the signal/noise ratio on these immunoblots is now much better than in the original submission. This has allowed us to revisit the issue of which acyltransferase is responsible for modifying this site. We now provide in new Figure 3a a C15:0-azide and C17:0-azide pulldown in the presence of ZDHHC7 knockdown, which shows that ZDHHC7 does not preferentially transfer C18:0 over C16:0. Upon ZDHHC7 knockdown, both drop mildly. This is consistent with previous reports that ZDHHC7 can use both C16:0 and C18:0 as substrates equally well (PMID 28167757). We now confirm this by providing in Figs. 3c and 3d evidence that ZDHHC7 binds both C15:0-azide and C17:0-azide on its active site cysteine. We thank the reviewer for having brought this up because we think it is important that we now clarified this issue.

For acylation of GNAI with C16:0, a screen has already been published (PMID 19001095) where all ZDHHC enzymes were tested and mainly ZDHHC3 and 7 were found to palmitoylate GNAI2, and to a lesser extent ZDHHC2 and ZDHHC21. Hence several ZDHHC enzymes appear to acylate GNAI redundantly. Indeed, a double knockdown of ZDHHC3+ZDHHC7 didn't show a strong drop in GNAI2/3 acylation (Fig. 3b). Hence multiple ZDHHC enzymes are likely redundantly acting on GNAI Cys3.

In sum, we find that GNAI palmitoylation vs oleoylation can be regulated by exposing cells to different levels of lipids, but not by altering activity of individual ZDHHC enzymes.

Additional experiments using ZDHHC7 loss-of-function mutants, or KO ZDHHC7 should also be performed to analyze the effects on GNAI signaling.

Since the new fatty-acid-azide pulldown assays which we provide in the revised manuscript, with a better signal/noise ratio than

the original blots, show that ZDHHC7 does not differentially affect GNAI palmitoylation versus oleoylation, we believe this issue is no longer relevant.

In addition, whether the deacylation rate would be affected by oleoylation?

We are not completely sure we understand this question. Is it whether the presence of C16:0 or C18:0 affects the rate of deacylation of GNAI proteins? Unfortunately we do not think we can reliably quantify deacylated protein, because the acylation is unstable as soon as the cells are lysed, so any de-acylated protein that we detect in mass spec or in the acyl-PEG exchange assay could be technical – e.g. Suppl. Fig. 2a. However, from Figure 2g-g' we know that the half-life of GNAI proteins is much longer than the rate at which they become acylated, meaning that most of the acylation is due to a cycle of deacylation and re-acylation. Hence the rate of acylation is determined by the rate of deacylation. In new Fig. 1b we see that the degree of labeling of GNAI proteins in the presence of 100 μ M C15:0-azide or 100 μ M C17:0-azide is the same, suggesting that the rate of deacylation of GNAI proteins is not strongly affected by the presence of the two different lipids.

2. The author claimed the level of C18:0 lipid exposed to cells effects acylation status of cys3 within GNAI. Do you think this modification alteration is directly controlled by the level of C18:0 or mediated by corresponding ZDHHCs? It is still unclear how this acylation change responds to the level of C18:0. Is this through the concentration of pal-CoA vs. oleoyl-CoA? The author should at least show that exogenous C18:0 could alter the intracellular fatty acyl-CoA composition.

We thank the reviewer for this question. It prompted us to do several additional experiments, resulting in an entirely new Figure 3 which now addresses this important point. As suggested by the reviewer, we quantified acyl-CoA levels and found that indeed exposure of cells to various lipids causes the corresponding acyl-CoA pool to increase (new Figure 3f). As shown in this figure, when cells are exposed to C16:0, the intracellular C16:0-CoA levels increase (in absolute terms, normalized to total phosphatidylcholine) whereas the levels of C18:0-CoA and C18:1-CoA do not. Conversely, exposing cells to C18:0 causes C18:0-CoA levels and C18:1-CoA levels to increase but not C16:0-CoA levels. Likewise, exposing cells to C18:1 causes C18:1-CoA levels to increase but not C16:0-CoA. As shown in new Figure 3e, we directly assayed the consequence of this on GNAI acylation. When cells are exposed to C16:0, this increases the proportion of GNAI that is palmitoylated whereas exposure of cells to C18:0 increases the proportion of GNAI that is oleoylated.

In contrast, as described in detail in the general comment above, we did not find any manipulation of ZDHHC proteins that alters the balance of GNAI palmitoylation versus oleoylation. This fits with the published literature showing that the ZDHHC enzymes are quite pleiotropic in terms of which substrates they can use (C16:0-CoA versus C18:0-CoA), and indeed ZDHHC7 seems to use both equally well (PMID 28167757).

In sum, we do think that the altered acylation is due to a shift in concentration of acyl-CoA species in the cell. (This also fits with the fact that the effect is happening very quickly - within 3 hours - which is too fast for transcriptional plus translational changes to lead to significant changes in ZDHHC protein levels.)

3. In figure 4b, it is not reasonable to study GNAI mediated association of EGFR-Gab1 by comparing IP results from WT and GNAI3 KO cells when input of EGFR-GFP and EGFR was not even (i.e. lane 4 vs 8). The same issue also occurred in Figure 4d.

(Now Figure 5b).

It appears that GNAI stabilizes EGFR protein because we consistently and reproducibly see less EGFR and EGFR-GFP in the GNAI KO cells. So this is a biological result, and not a technical one.

Importantly, however, we took this into account by equalizing the amount of EGFR in the IP (please see arrows in Reviewer Figure 5 below). This is the relevant parameter for assaying how much Gab1 is binding to EGFR, not the amount of EGFR in the input. Indeed, in the quantification of panel b we are showing levels of co-IPed Gab1 normalized to IPed EGFR. We realize this was not very clear in the original submission, and have now added this explicitly to the figure legend.

4. To confirm C18:0 affects EGFR signaling via GNAI proteins, the author knockout GNAI3 to test whether C18:0 would impact Gab1 recruitment (Figure 4d). In GNAI3 KO cells, however, the recruitment of this docking proteins is too low to respond to C18:0 treatment. Overexpression of ZDHHC7 or GNAI3 would provide a stronger basal level of Gab1 recruitment, thereby mirroring the effects of oleate treatment.

As suggested, we now show in new Figure 5d that overexpression of GNAI3 rescues the reduced Gab1 recruitment caused by C18:0.

(Please note that oleate treatment is reducing the amount of GNAI in DRMs, not increasing it. Therefore, GNAI3 overexpression would not be expected to phenocopy the oleate treatment, but rather to rescue the phenotype, as we see in Fig. 5d).

5. Palmitoylation of GNAI proteins facilitates its partition into DRMS. However, treatment of C16:0 to cells did not show increasing P-AKT level when co-treated with EGF. Even more, without EGF stimulation, C16:0 treatment also decreased the EGFR signaling (Figure 5C). One possible reason is that effects of C16:0 is miscellaneous. Therefore, the author should identify transferases for palmitoylation or oleoylation of GNAI at cys3. Then, overexpression of corresponding ZDHHCs to confirm acylation of this sites indeed regulates downstream EGFR signaling.

In new Fig. 3e we analyzed the acylation of GNAI3-GFP in cells treated with either C16:0 or C18:0. Compared to control cells, C16:0 does shift the ratio of GNAI palmitoylation vs. oleoylation towards palmitoylation, just like C18:0 shifts the ratio towards oleoylation. Therefore, also exogenous C16:0 affects the acylation of GNAI3 in the cell, as expected.

However, under control conditions, we do not think the levels of GNAI3 in the DRMs are limiting for EGFR signaling. This can be seen in Fig. 5d where GNAI3 overexpression in the control condition (without C18:0) does not increase the stability of EGFR protein or the recruitment of Gab1 upon EGF stimulation. Hence it appears that in control conditions sufficient GNAI protein is already present in the DRMs to allow maximal EGFR signaling. As a consequence, although C16:0 treatment of cells increases the proportion of GNAI that is palmitoylated, this does not have an effect on EGFR signaling compared to the control condition. This explains why the C16:0 treatment does not cause increased pAkt levels.

6. The author should show whether C18:0 promotes cell proliferation through GNAI oleoylation. Although Suppl. Figure 6a-b suggested GNAI 1 and 3 was associated with cell proliferation, more solid evidence is required to confirm such cell growth is through GNAI oleoylation when treated with C18:0, at least in part. For example, will the

sensitivity to C18:0 treatment alter in the context of GNAI KO or overexpression? Will exogenous GNAI restore the sensitivity in GNAI-depleted cells?

We assume the purpose of the proposed experiment restoring the reduced proliferation rate of GNAI-depleted cells with exogenous GNAI is to show that the phenotype is on-target. In Suppl. Fig. 7a we show the phenotype using siRNAs and in Suppl. Fig. 7b using a CRISPR knockout line. Since these are two completely independent methods of depleting GNAI3, it shows that the phenotype is not an off-target effect.

Please note that we do not want to claim the effect of C18:0 is going entirely through GNAI3. Indeed, we wrote "Since exposure of cells to C18:0 leads to stearoylation of several different proteins, the effect of C18:0 on cell proliferation is likely in part via GNAI proteins and in part via other mechanisms." That said, Akt activity is known to affect cell proliferation, hence it is very likely that it is contributing. The difficulty in addressing the reviewer's point is that the effect of C18:0 on proliferation is mild, so dissecting how much of this is GNAI3 dependent is technically challenging.

7. In vivo study is suggested to show C16:0 and C18:0 modification of GNAI3 influence competitively tumor growth. It is also of interest to link different diets wherein the ratio of C16:0 and C18:0 vary to tumor growth in vivo.

This is a good point, and has been already done (PMID 19267249, 24832758, 7214328, 3689663), showing that, consistent with our results, dietary stearate reduces tumor growth and overall tumor burden.

Minor issues:

1. The author tested silencing of several ZDHHC palmitoyltransferases on the acylation level of GNAI2 and stated that ZDHHC7 is the main enzyme responsible for modifying GNAI2 in page 10. However, in Figure 1c, silencing of ZDHHC7 didn't reduce the acylation level of GNAI2. Technical issue exists within the group of ZDHHC7 knockdown, since no C15:0-azide or C17:0-azide treatment has strong signal (lane 10), the experiment should be repeated. Furthermore, it would be of higher scientific value to screen all the ZDHHCs to find the strongest ZDHHC responsible for GNAs. Since the basal level of GNAs fatty acylation is low, it's would be interesting to overexpress ZDHHCs to look at fatty acylation of GNAs. At the same time, any preferences of ZDHHCs among GNAI1, GNAI2 and GNAI3?

We thank the reviewer for pointing this out. As a consequence, we optimized the click-it pulldowns and we now have a much better

signal/noise ratio compared to before (new Fig 1b, Fig. 3a-d). As detailed in the General Comment above, we no longer believe ZDHHC7 specifically affects oleoylation of the GNAI proteins. Rather, modification with both C15:0-azide and C17:0-azide are mildly affected (Fig. 3a). This fits with the fact that ZDHHC7 can use both C16:0 and C18:0 as substrates. Furthermore, a systematic screen of all ZDHHC enzymes on GNAI proteins was already previously done (PMID 19001095), and the main enzymes acylating GNAs were found to be ZDHHC3 and 7, with ZDHHC2 and 21 contributing to a much lesser extent. Nonetheless, in vivo, double knockdown of ZDHHC3 and 7 doesn't strongly reduce GNAI acylation, indicating that multiple ZDHHC enzymes are working redundantly (Fig. 3b). Also, as suggested by the reviewer, we tried ZDHHC7 overexpression, but this also has no effect on GNAI acylation (Fig. 3c) indicating that ZDHHC activity levels are not limiting for GNAI acylation in vivo. Altogether, as discussed in the General Comment above, this indicates that the lipid levels and not the ZDHHC enzymes are regulating GNAI acylation.

2. In Figure 3a, a membrane marker should be included for staining. Since the author potentiates the effects of fatty acylation on GNAs membrane localization, it would be better to treat the cells w/o fatty acids to look at the change of GNAI membrane localization by IF.

We now include as Suppl. Figure 3a and Suppl. Figure 4a cell stainings including a membrane marker. As suggested by the reviewer, Supplementary Figure 4a also shows cells treated with medium containing delipidated serum (i.e. cells treated without fatty acids) and this has no effect on the membrane localization of the GNAI proteins (which is expected from the fact that MCF7 cells have FASN and hence can biosynthesize their own C16:0.)

3. In Figure 3c, the input level of GNAI3 is higher in BSA group, it would be suspicious that C16:0 and C18:0 reduces the fractions of DRMs was due to the reduced total protein level.

(Now Fig. 4c).

Thank you for raising this issue, which we have now fixed: The BSA blot and the C18:0 blot are on separate membranes. Hence, although we controlled everything as much as possible, differences in loading levels and exposure times of the blots will influence the absolute level of the signal seen on the figure. For this reason, the quantification shown in Fig 4c of GNAI3 levels in DRMs was normalized to total GNAI3. We did this by quantifying the GNAI3 band intensity for all the fractions, and then calculating the intensity of the DRM fractions divided

by the total of all fractions. (The input was not used for this calculation). Hence this normalizes for mild changes in the parameters mentioned above. We have now added this to the figure legend.

Another important point is that the immunoblot images were acquired with a Chemidoc Imaging system which has a dynamic range of 65536 intensity levels per pixel, whereas the images shown on the screen only have a dynamic range of 255 greyscale levels. Hence only a part of the total dynamic range of the data is shown when exporting the image. We have now exported the BSA GNAI3 blot differently so that the input samples in the BSA condition have a similar intensity to the input samples in the C18:0 condition (Fig. 4c). Please note this does not affect the quantifications because the quantifications are done on the underlying data in the Chemidoc system, and not on the exported image.

Finally, the upper band in the GNAI3 blot is non-specific. Somehow this got lost during figure preparation. We have now fixed this by adding an arrowhead with "n.s." to the figure.

4. In Figure 4b and 4c, GNAI3 KO reduced the total protein level of EGFR (input), the reduced pulldown of EGFR and Gab1 may due to the reduced total protein level. It would be interesting to overexpress GNAI3 to check the response to EGF treatment.

If we understand correctly, this is the same issue as raised by the reviewer in Major Point 3. Please see the response to Major Point 3 above.

5. In Figure 5a, silencing of GNAI1 is not confirmed.

Please note that we explained this in detail when presenting the characterization of the antibodies in Suppl. Figure 1. In brief, GNAI1 and GNAI3 proteins are extremely similar in sequence so that the antibodies for GNAI1 and GNAI3 cross-react and detect both proteins. This can be seen from the overexpression study presented in Suppl. Figure 1a. Since MCF7 cells express much more GNAI3 than GNAI1 (Suppl. Figure. 1d), then both GNAI1 and GNAI3 antibodies detect GNAI3: there is a drop in signal when GNAI3 is knocked down, but not when GNAI1 is knocked down, because in this case they still detect the GNAI3 that is present (Suppl. Figure 1b). Hence it is not possible to quantify the drop in GNAI1 protein levels caused by the GNAI1 siRNA.

6. In Figure 5c, C18:0 treatment has no effects on EGF induced phosphorylation of AKT, the statement in page 17 is not right.

(Now Fig. 6c)

The drop is mild in magnitude when normalized to total Akt. It is more visually apparent on the p-Ser473 site than the pThr308 site. For this reason, we added a quantification below the immunoblot in Figure 6c' (reproduced as Reviewer Figure 6 below) which shows that over multiple biological replicates there is a 40% drop in both pAkt(Thr308) and pAkt(Ser473) when comparing the C18:0 treated cells to the BSA treated control cells in the presence of EGF (see red arrows below). This is highly statistically significant ($p=0.0002$ for P-Thr308 and $p=0.0005$ for P-Ser473) and consistent across replicates.

7. In page 13, no in vivo experiment was performed in the current study, the author should correct the statement.

We thank the reviewer for pointing out that the term 'in vivo' can mean different things to different people – some use it to contrast to 'in vitro', in which case cells are 'in vivo'. Others use it to contrast cell culture to animal models. In our case, the sentence is drawing a conclusion from the results presented in the paragraph, which are in live cells, which is what we meant with 'in vivo'. To avoid this possible misunderstanding, we have now replaced this with 'in cells' to make it more clear.

Technical issues:

In supplemental Figure 1a, GNAI3 antibody (ab173527) is not working for detection of GNAI3 but GNAI1. GNAI1 antibody (ab14015) only detects GNAI1, marginally detects GNAI3. Thus, the description on page 9 should be revised. At the same time, no antibody is working for GNAI3, it would be hard to detect endogenous GNAI3 and confirm the silencing effects of siRNA. The author should seek another GNAI3 antibody.

We believe there is a misunderstanding. In Suppl. Figure 1a, by looking at the V5-tag blot, one sees that GNAI3-V5 is not overexpressed as strongly as GNAI1-V5. For this reason, the GNAI3 antibody gives a stronger signal in the lanes with overexpressed GNAI1 compared to overexpressed GNAI3. Nonetheless, by comparing the GNAI3 lanes (10-12) to the negative control lanes (1-3), one clearly sees an additional band appearing, meaning that the GNAI3 antibody does detect overexpressed GNAI3. The same is true for the GNAI1 antibody, which gives a stronger signal in lanes 4-6 than lanes 10-12. This is because both GNAI1 and GNAI3 antibodies detect both proteins given that they are almost identical in sequence. Regarding detection of endogenous protein, Suppl. Figure 1b shows that the band detected by the GNAI3 and the GNAI1 antibodies goes away upon GNAI3 knockdown, indicating that it is indeed GNAI3 that they both detect. This is because GNAI3 is much more strongly expressed in MCF7 cells than GNAI1 (Suppl. Fig. 1d). That said, as the reviewer points out, the "GNAI1" antibody gives overall a stronger signal than the "GNAI3" antibody, and we often used the GNAI1 antibody to detect endogenous GNAI3.

In Figure 1b, the signal of GNAI2 and GNAI3 pulldown is quite weak, the author should improve the quality of the blot.

We have optimized the click-iT pulldown and repeated the experiment and have now replaced Figure 1b with new Western blots.

Reviewer #3:

The manuscript by Nuskova et al claimed that stearic or oleic acid blunts EGFR and AKT signaling caused by the oleoylation of a cystein residue on GNAI. They proposed a mechanism by which oleoylation of GNAI leads to the exclusion of GNAI with EGFR in the detergent resistant membrane (DRM), in contrast to palmitoylation or stearylation on the same residue. The study is interesting and likely to shed new light into a posttranslational modification of a protein by different fatty acid may affect the biological activities.

We thank the reviewer for the positive assessment of the interest and novelty of our study.

However, there are some major concerns regarding this study and their final conclusions. First of all, if the oleoylation of C3 on GNAI is responsible for the potential beneficial effects of anti-tumor properties, one would argue that oleic acid is better than stearic acid in this regard. Accordingly, there are more human data to link C18:1 to health benefits than stearic acid. As the authors are aware that there are more negative health effect for stearic acid in humans than the "anti-tumor effects".

We thank the reviewer for pointing this out. We agree. We mainly wanted to contrast the two saturated fatty acids C16:0 and C18:0 and there are data (including controlled animal experiments) showing that C18:0 is better than C16:0 in terms of tumor development. That said, C18:1 is even healthier. We now mention C18:1 in the Abstract and in the Discussion.

Secondly, it remains to be clearly defined whether C18:1 oleoylation on C3 occurs on the protein or only oleic acid can be used as a substrate. Although the author attempted to use a desaturase inhibitor in their experiments, it is still inconclusive to tease this out. An alternative would be to carefully monitor the fatty acid composition by GC-MS in the cell, either in free form or esterified to see if C18:0 has to be desaturated before it is acylated onto the protein.

We politely disagree with the reviewer that the results are inconclusive. Our data show that either C18:0 or C18:1 can be added to the GNAI proteins. When we add ^{13}C -labeled C18:0 to cells in the presence of the desaturase inhibitor, we clearly see the ^{13}C -labeled C18:0 esterified on GNAI3 (Fig. 2e). Hence C18:0 can be added to GNAI3. Likewise, if we add heavy C18:1 to cells, we see the heavy C18:1 esterified on GNAI3 (Fig. 2f). Hence both routes are possible.

Admittedly, what was not clear is the relative flux ratio of these two routes in cells. As suggested, we have now quantified acyl-CoA levels in cells upon addition of C16:0, C18:0 or C18:1 to cells (new Fig. 3f). This shows that exposure of cells to C18:0 increases both C18:0-CoA and C18:1-CoA levels. In fact, in absolute terms, upon exposure to C18:0, the increase in C18:1-CoA levels (by 0.73×10^3) is larger than the increase in C18:0-CoA levels (0.19×10^3). In contrast, exposure of cells to C18:1 increases only C18:1-CoA levels. This indicates that the main route is for C18:0 to first be desaturated to C18:1-CoA and then added to the GNAI proteins. This agrees with our previous data using the acyl-CoA desaturase inhibitor MF-438 which essentially abolished the C18:1 acylation of GNAI3 (Fig. 2e). We have updated the manuscript accordingly.

With that said, it is still unclear how the the cells use this mechanism in vivo. C18:0 or C16:0 is the predominant fatty acid in the cell membrane. Why they have to use the

exogenous fatty acid for this purpose? In other words, it is hard to imagine that the benefits of C18:0 is only observed after dietary supplementation as in the study cited in ref.14 when our body have plenty of C18:0.

We agree with the reviewer that the finding from our clinical study in Ref. 14 is unexpected and exciting. It is exactly for the reason mentioned by the reviewer that we performed this clinical study. Since cells can endogenously produce both C16:0 (by FASN) and C18:0 (via Elovl6), it is not predictable *a priori* whether cells would be sensitive to the levels of exogenous lipids that they experience. This, however, was exactly the finding from the clinical study – within 3 hours of ingesting C18:0, the cells in the human body activate the signaling pathway involving stearoylation of TfR1. This result was highly statistically significant, and consistent between both healthy subjects and diabetic patients. Hence the exogenous pool of lipids seems to affect cell physiology differently from the endogenous pool. This can either be due to the fact that cells have only a limited capacity to synthesize C16:0 and C18:0, and indeed these lipids are mainly used for cell membrane biosynthesis, or to the fact that the lipids are entering the cells from different routes.

We now also include in the revised manuscript two sets of data that fit with this. Firstly, in Fig. 3f we quantified levels of acyl-CoA from cells treated with exogenous C16:0, C18:0 or C18:1, and indeed we see that the corresponding intracellular acyl-CoA levels respond to the exogenous fatty acids. This indicates that endogenous fatty acid biosynthesis does not saturate intracellular acyl-CoA levels. Secondly, we quantified the levels of GNAI palmitoylation and oleoylation in cells exposed to exogenous C16:0 or C18:0 and see that indeed exposure to C16:0 increases the proportion of GNAI palmitoylation whereas exposure to C18:0 increases the proportion of GNAI oleoylation (Fig. 3e).

In sum, all the data, both in people and in cell culture, indicate that human cells are responding to exogenous lipids despite their endogenous synthesis.

Below are some minor concerns:

1. The concentrations of azido fatty acids in the click experiments were not shown.

We thank the reviewer for pointing this out. The concentration (100 μ M) was indicated in some figure legends, but not all. We have now added it to all the figure legends.

2. If C18:1 modification has such a dramatic effect, one would expect similar effect on C16:1 modification at the same site. Similar modifications is observed in Fig.2C.

We agree with the reviewer that it is possible that C16:1 would have a similar effect as C18:1. However, the main saturation/desaturation shift that seems to occur is between C16:0 and C18:1, and not between C16:0 and C16:1. This is for several reasons:

- Cells appear not to use C16:1 much for modifying GNAI proteins. The C16:1 peak in the panels “+¹³C18:0” and “+¹³C18:1” of Figures 2d & 2f is small compared to the C18:1 peak.
- The heights of the C16:0 and C16:1 peaks are coupled to each other, probably by the intrinsic ratio of C16:0 desaturation in the cell. C16:1 is not a predominant dietary lipid.
- In contrast the C18:0 seems to mainly be converted to C18:1 acylation on GNAIs. Hence exposure of cells to either C18:0 or C18:1 shifts the balance between saturated C16:0 and desaturated C18:1.

We have added a comment to the Discussion that C16:1 would be predicted to have similar effects.

3. Figure 2 is critical for the identification of these novel modifications. Please label m/z in the upper panel in Fig.2C. Zoom-in the region between 1800 to 2000 would really help reader to see the modifications. In the text, the authors mentioned a broad peak in these spectra but they are not clear.

As suggested, we have now labeled m/z in Fig. 2c-f and provide a zoom-in of the region between 1800-2000 on the right side of Fig. 2c-f. (We moved the fragmentation spectra to Suppl. Fig. 2b)

4. The fragmentation in Fig.2D did not directly reflect the nature of the modification. These modifications are indirectly derived from the parent ion and y ions.

We agree with the reviewer that the fragmentation data in old Fig. 2D were simply shown to verify the identity of the peptide. Because of their lower importance for the investigation of acylation, we moved them into supplements (Suppl. Figure 2b)

Did the myristoylation of the G occur endogenously?

Yes – the myristoylation is endogenous.

What is the broad peak at 1750 shown up in panel 2, 3, and 5 of Fig. 2D?

The broad peak at 1750 m/z is not actually a peak but noise located at the boundary between the parent ion selection area and the measured daughter ions area. It is a technical drawback (blind spot) that typically appears when the "lift" MALDI TOF-TOF fragmentation is performed. The weaker the parent ion, the bigger the false splash we observe. That is why those splashes are not equal in different MS/MS spectra. This zone (close to the parent ion m/z value) cannot be interpreted scientifically.

5. In the first graph on P11, "In contrast, palmitoylation of GNAI2 was retained upon knockdown of either ZDHHC3 or ZDHHC7.." ZDHHC3 or ZDHH6?

We thank the reviewer for pointing this out. Actually, it is retained upon knockdown of any of the three (ZDHHC3, 6 or 7). In this revised manuscript, we have significantly improved the signal/noise ratio of the click-it pulldown assays (new Figs. 1b, 3a-d). These data now show that acylation of the GNAI proteins is performed by multiple ZDHHC enzymes in parallel. A previous systematic screen of all ZDHHC enzymes on GNAI proteins (PMID 19001095) showed that the main enzymes acylating GNAI s are ZDHHC3 and 7, with ZDHHC2 and 21 contributing to a much lesser extent. Nonetheless, double knockdown of ZDHHC3 and 7 doesn't strongly reduce GNAI acylation, indicating that multiple ZDHHC enzymes are working redundantly (Fig. 3b). Furthermore, we find that overexpression of ZDHHC7 does not increase GNAI acylation, meaning that ZDHHC activity is not limiting (Fig. 3c). Instead, acylation of the GNAI proteins is strongly affected by exposure of the cells to various lipids. We now provide new data in Fig. 3e showing that when cells are exposed to C16:0, palmitoylation of GNAI goes up in proportion to oleoylation, whereas exposure to C18:0 causes GNAI oleoylation to increase. In sum, the differential acylation of the GNAI proteins is mainly driven by substrate availability (i.e. exposure to lipids) rather than ZDHHC activity. We have updated the manuscript accordingly.

6. Where does oleoylation occur in the cells? Does it happen before the protein transported to the membrane or on the membrane?

According to the literature, these acyl transferases are present on the surface of the ER or Golgi, hence it most likely happens before it is transported to the membrane.

REVIEWERS' COMMENTS

Reviewer #1 (Remarks to the Author):

The authors have done an extensive re-write and revision of the previous manuscript and have addressed nearly all of my concerns. In particular, they have considerably improved the quality of their labeling experiments and blots, and as a result, the statistical significance of the data is more convincing.

Reviewer #2 (Remarks to the Author):

The revised manuscript has been greatly improved. The authors have performed additional experiments to address my previous concerns, and new findings have been presented.

Minor comments: The new data showed that ZDHHC3/7 are not major regulators of GNAI acylation, and KD these enzymes has little impact on GNAI acylation. These results are different from previous report, and might suggest that additional enzymes or mechanisms are involved. The author should discuss these discrepancies more thoroughly.

Reviewer #3 (Remarks to the Author):

The authors addressed all my concerns

Reviewer #1 (Remarks to the Author):

The authors have done an extensive re-write and revision of the previous manuscript and have addressed nearly all of my concerns. In particular, they have considerably improved the quality of their labeling experiments and blots, and as a result, the statistical significance of the data is more convincing.

Thank you !

Reviewer #2 (Remarks to the Author):

The revised manuscript has been greatly improved. The authors have performed additional experiments to address my previous concerns, and new findings have been presented.

Minor comments: The new data showed that ZDHHC3/7 are not major regulators of GNAI acylation, and KD these enzymes has little impact on GNAI acylation. These results are different from previous report, and might suggest that additional enzymes or mechanisms are involved. The author should discuss these discrepancies more thoroughly.

We have added the following to the discussion:

“A previous study found that GNAI proteins can be acylated by ZDHHC3 and ZDHHC7 when co-transfected in HEK293T cells, with a smaller contribution from ZDHHC2 and ZDHHC21 51. The fact that knockdown of ZDHHC3 and 7 only had a minor impact on GNAI acylation (Fig. 3) suggests that endogenous ZDHHC2 and ZDHHC21 may also contribute to GNAI acylation in vivo, and that multiple enzymes are acting redundantly in this regard. Altogether, our data suggest that the relative acylation of GNAI3 Cys3 with C16:0 or C18:1 does not depend on changes in enzyme activity, but reflects instead the balance in fatty acids to which a cell is exposed.”

Reviewer #3 (Remarks to the Author):

The authors addressed all my concerns

Thank you !